# Potential role of developmental experience in the emergence of the parvo-magno distinction

Marin Vogelsang [1], Lukas Vogelsang [1] ✉, Gordon Pipa[2], Sidney Diamond[1] & Pawan Sinha[1]

While the division of the early visual pathway into parvo- and magnocellular systems with distinct response properties has long been established as a prominent organizing principle in mammalian visual systems, the factors that lead to its emergence remain unclear. Here, we provide a potential account of this emergence based on early sensory development. Specifically, we propose that the temporal confluence in the developmental progression of spatial frequency and chromatic sensitivities may significantly shape corresponding neuronal response properties characteristic of this division. Receptive field analyses of deep networks trained on developmentally inspired 'biomimetic' protocols support this proposal in both the spatial and temporal domain. Further, biomimetic training induces a more human-like bias towards global shape processing, potentially driven by magnocellular units. These results have implications for the emergence of a key aspect of visual pathway organization and applied relevance for the design of training procedures for computational vision systems.

Cells in the early visual pathway in mammalian brains can be broadly segregated into two groups: magnocellular and parvocellular[1,2]. Two key characteristics of this division, already strongly evident by the lateral geniculate nucleus (LGN) and originating at the level of retinal ganglion cells[3], are related to color[4–7] and spatial frequency sensitivities[8–10]. Magno units exhibit receptive fields (RFs) markedly larger than those of parvo cells and are mostly achromatic, while most parvo cells are tuned to color content. Thus, the magnocellular group exhibits both low spatial frequency and chromatic sensitivity, and the parvocellular group exhibits both high spatial frequency and chromatic tuning.

In addition to these spatial characteristics, the two pathways also differ in their temporal response properties. Magno cells are known to exhibit faster conductance[5,11] and higher temporal frequency sensitivity[8–10]. Specifically, the temporal frequencies eliciting maximal activation were found to be approximately twice as high in magnocellular neurons in the monkey LGN, relative to parvocellular ones[8,9]. The magnocellular system also includes a greater proportion of biphasic cells, which reverse their phase responses over time and thereby enable the encoding of rapid temporal transients[12].

Drawing on these spatial and temporal characteristics from neurophysiology, along with early psychophysical and clinical studies[13,14], the magnocellular system has typically been assigned a more global function in spatial organization[1]. Its role has also been likened to the processing of Gestalt cues for object understanding and linking properties[1,13]. In contrast,

the parvocellular system has been implicated in the analysis of greater spatial detail[1]. Recent human neuroimaging studies have more directly attested to the involvement of areas in the dorsal visual pathway, which are driven by strong magnocellular input, to global processing[15,16]. For instance, dorsal areas of the visual system were shown to be crucial for the analysis of the broader spatial arrangement and global shape of object parts[17] as well as for the configural processing of faces[18]. Disrupting processing in these areas using transcranial magnetic stimulation was furthermore shown to lead to a specific impairment of global, not local, processing[19]. Thus, even though both parvo- and magnocellular systems are likely to contribute to complex behavior, and their individual contributions cannot be entirely disentangled, these studies speak to the significance of the magnocellular system in global processing.

While the anatomical, physiological, and psychophysical division between the two pathways has been widely accepted, details of their emergence from the molecular and cellular processes of early development into their functional use in experiential vision are not yet clearly established. Here, we propose and computationally test an account based on early developmental trajectories of sensory experience. Specifically, we examine whether the tuning of some units (magnocellular-like) to both low spatial frequency and low color content could potentially be due to the degraded color and frequency characteristics of visual inputs early in life, with parvocellular-like units being more optimized for processing high-fidelity inputs that become available later in development. To this end, we test here

[1]Department of Brain and Cognitive Sciences, Massachusetts Institute of Technology, Cambridge, MA, USA. [2]Institute of Cognitive Science, Osnabrueck University, Osnabrueck, Germany. ✉e-mail: lvogelsa@mit.edu

whether a relatively generic deep convolutional neural network, when exposed to a developmentally inspired trajectory of training inputs, is able to recapitulate key aspects of parvo- and magnocellular organization in terms of receptive field properties and functional correlates in terms of global shape processing.

Providing some background for this developmental proposal, as is the case with many aspects of human perceptual development, color sensitivity[20,21] and visual acuity[22,23] mature over the months following birth from limited to proficient. The underlying factors are believed to be the maturation of retinal photoreceptor morphology and transductional efficiency, as well as elaboration of circuits in the retina and cortex[24–27]. Recent computational investigations have begun exploring potential benefits conferred by such early degradations, although, thus far, focusing on individual visual dimensions (e.g., acuity or color) in isolation.

In the domain of visual acuity, training deep convolutional neural networks with inputs that transition from blurry to non-blurry was shown to lead to first-layer receptive fields tuned to lower spatial frequencies, effectively integrating information over more extended areas of the input image[28]. This training progression, compared to several non-developmental ones, also yielded the best generalization performance across varying blur levels, suggesting potential benefits of commencing training with blurry inputs, at least in the domain of face recognition[28,29]. Although without an explicit developmental motivation, training a deep network with various levels of blurred inputs (including strong blur) was recently shown to also result in greater similarity with neural responses in macaque visual cortices, greater robustness to image perturbations, and an increased bias to classify images based on global shape rather than local features[30]. When training with a mix of blurred and non-blurred images or when transitioning from blurred to non-blurred inputs, an increased shape bias, albeit a more modest one, had also been reported[31]. Separately, in the domain of color vision, a deep network training progression transitioning from achromatic to full-color inputs was found to result in greater robustness to later color removal or chromatic changes as well as to more luminance-based receptive fields[32]. Together, these studies highlight the potential benefits of sensory degradations early in life – a proposal we have referred to as the 'adaptive initial degradation' hypothesis[33]. While this past work underscores how initial degradations might be advantageous for the developing system, it has, thus far, focused on isolated visual dimensions.

In biological development, however, sensory functions such as acuity and color sensitivity mature concurrently, albeit on slightly different timescales. The perspective that drives this investigation is that the joint developmental progression of visual acuity and color sensitivity could lead to additional key insights, including how these two dimensions are encoded in the neural substrates. Specifically, the start of visual experience is accompanied by poor acuity and low color sensitivity. Hence, the response properties of some cells emerging at this time could come to reflect both of these constrained signals, resulting in units being tuned to low spatial frequency and low color content. As described above, these would correspond to the magnocellular system. As development progresses, higher acuity and richer color information become available and could be incorporated into corresponding neuronal response properties. These would be more akin to the parvocellular system. (It is worth prefacing here that the well-separated parvo and magno layers in the LGN begin to mix in the visual cortex[7]. Consequently, two subpopulations of parvocellular units have been reported in the literature, both of which are color-sensitive to some extent: those exhibiting higher spatial frequency selectivity and typically strong orientation tuning ('interblobs'), as well as those exhibiting lower spatial frequency selectivity and mostly low orientation tuning ('blobs')[7]. Given these subspecializations that are plausible in the context of richer visual inputs, we would expect a greater heterogeneity of receptive field tuning in the parvocellular system, relative to the magnocellular one.)

To probe this developmental account of the emergence of the parvo- and magnocellular systems, we conducted simulations with deep convolutional neural networks. While these powerful networks are not without their limitations as computational models of the biological system[34–36], they are among the best to predict human behavior and neural responses[37–39]. Moreover, they offer a framework for scientific exploration and the testing of neuroscientific hypotheses[40,41]. Of particular relevance to us, they provide a systematic methodology for directly probing the consequences of deliberate manipulations of early sensory inputs[42]. Unlike in human participants, where such manipulations are not feasible for clear ethical reasons, we can expose a deep network to different temporal progressions of inputs and analyze the effect on learned representations and performance outcomes. Some of these progressions can be chosen to recapitulate aspects of typical development, while others could serve as non-developmental controls. In the present study, this approach allows us to examine the differently trained networks' receptive fields in the early stages of their architecture in terms of their color and spatial frequency tuning. In addition, it permits us to probe the functional roles of these receptive fields for the network's behavior in terms of reliance on global shape and local textural features.

To operationalize this approach, we trained a (slightly adapted) AlexNet[43] on the ImageNet database[44] using two separate temporal stimulus progressions, or 'training regimens', of interest ('standard' and 'biomimetic'):

i. In the 'standard' regimen, as a non-developmental control, we trained our network on high-resolution, full-color images for the entire training duration comprising 200 epochs.
ii. In the developmentally-inspired 'biomimetic' regimen, we trained the network on reduced resolution, achromatic images for the first 100 epochs, and on high-resolution, full-color images for the subsequent 100 epochs.

To examine the generalizability of our results, these comparisons were carried out across several network settings, which are illustrated in Supplementary Fig. 1. In 'setting 1', for which results are reported in the main manuscript, we adapted the AlexNet to feature larger first-layer filters (increased from $11 \times 11$ to $22 \times 22$ pixels) in order to avoid restricting the types of receptive field structures to be learned and to allow for the extraction of frequency-based metrics with higher resolution. We also reduced the number of receptive fields from 96 to 48 as a precautionary measure to increase the likelihood of all receptive fields learning clear structures that can be meaningfully quantified with metrics. In addition, we also report results with the full set of 96 receptive fields (size increased to $22 \times 22$ pixels) ('setting 2'), with the full set of 96 normally-sized (i.e., $11 \times 11$ pixel) receptive fields ('setting 3'), with only half of the training epochs ('setting 4'), and with a decreasing learning rate ('setting 5'). Based on setting 1, we also trained the network on three additional biomimetic regimens ('biomimetic v2-v4') (see Methods). Training of each network was thereby repeated with five random initializations. While for receptive field analyses, the results are depicted separately for each training run, for all other analyses, the results of the five different runs are summarized in a single figure. More details of training and analysis can be found in the Methods section. Notably, the key results reported in this paper generalize well across training runs and different settings tested.

## Results
### Learned receptive field structures

Figure 1A, B depicts the learned convolutional filters (henceforth, 'receptive fields') in the first layer of our network following training with the 'standard' and the developmentally inspired 'biomimetic' regimen. Shown here are the results of the first training run; outcomes of all five runs with different random initializations are shown in Supplementary Fig. 2. As is evident by visual inspection, and as quantified in Fig. 1C, D showing the distribution of individual receptive fields' chromatic and spatial frequency tuning, training with the biomimetic regimen results in receptive fields that are tuned less to high spatial frequency and chromatic content. This effect highlights the significance of spatially extended and luminance-based receptive fields instantiated during the first half of training. Notably, these differences in RF sensitivity distributions persist notwithstanding the second half of training with high spatial resolution and high chromatic content stimuli.

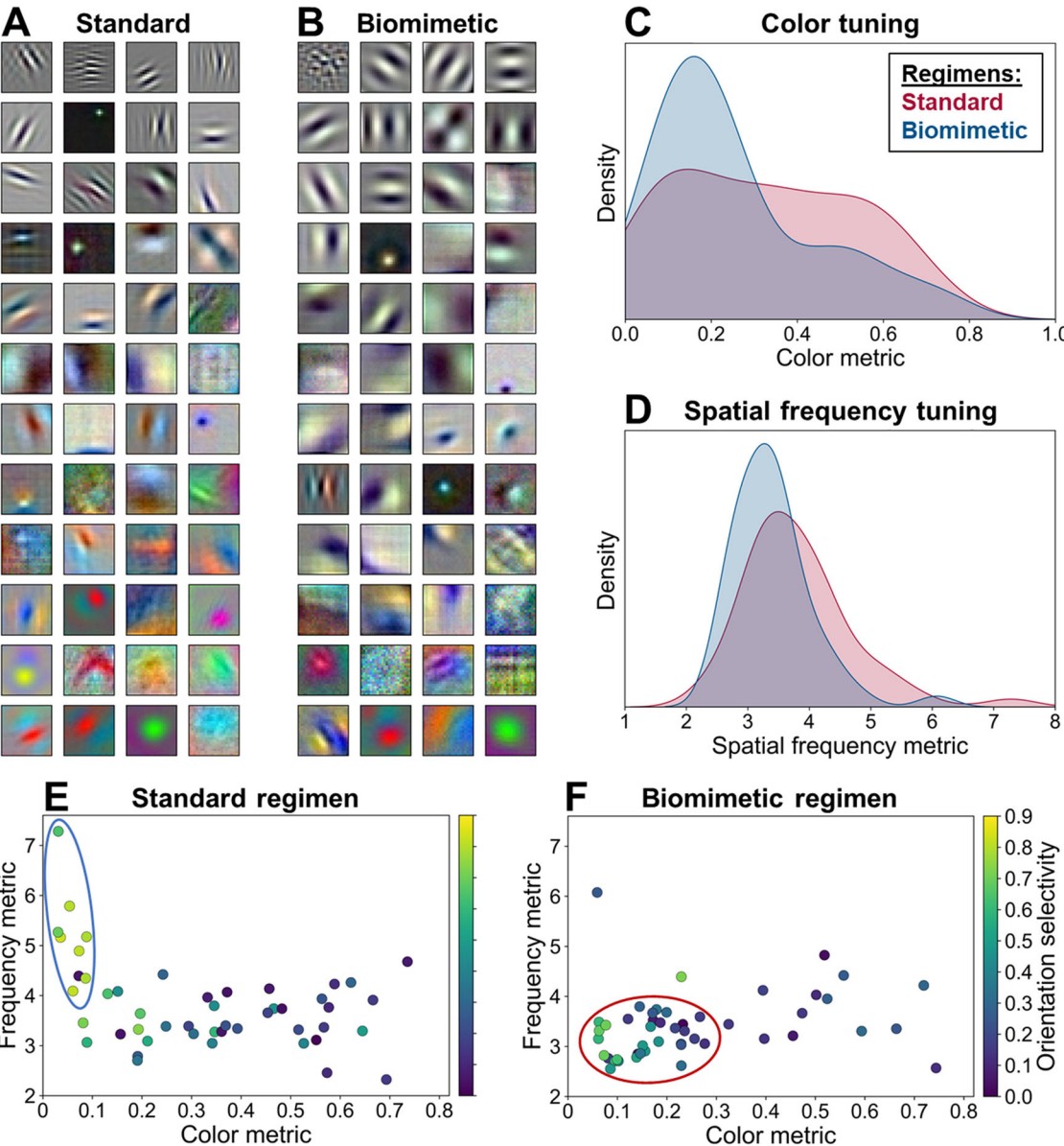

**Fig. 1 | Results of receptive field analysis.** First-layer RFs following training with the standard (**A**) and biomimetic (**B**) regimen. Color (**C**) and spatial frequency (**D**) distributions of individual RFs of both models. Scatter plots depicting the joint frequency and color coding of individual RFs following training with the standard (**E**) and biomimetic regimen (**F**). Depicted here are results obtained with the first training run; qualitatively similar outcomes of five training runs with different random initializations are shown in Supplementary Fig. 2.

Of particular relevance to the parvo-magno distinction, Fig. 1E, F depict the joint distribution of frequency and color coding of individual receptive fields, revealing marked differences between the two regimens. In the 'standard' network, no clear relationship between the two attributes is evident, except for the existence of a few high-frequency achromatic receptive fields (Fig. 1E, blue ellipse). These, however, do not map onto either the magnocellular or the parvocellular group of neural units but rather reveal an anti-correlation between the two attributes. In the biomimetic network, by contrast, we observe the presence of many receptive fields that exhibit magnocellular-like characteristics in terms of their tuning to both low frequency and low chromatic content (Fig. 1F, red ellipse). The strong joint coding observed in these units is especially noteworthy given that it is not a necessary consequence of both color and spatial frequency distributions being tuned to lower values; they nevertheless could have emerged to be independent.

While the units of the biomimetic model tuned to high spatial frequency and/or high chromatic content exhibit a greater heterogeneity than the magnocellular-like ones (Fig. 1F, dots outside of the red ellipse), this observation is expected. As mentioned in the introduction, two subpopulations of parvocellular units have been reported in the biological system: interblobs, exhibiting high spatial frequency selectivity and strong orientation tuning, and blobs, exhibiting lower spatial frequency selectivity and low orientation tuning[7]. A closer examination of Fig. 1F (and corresponding panels in Supplementary Figs. 2–12), where colors code for a given unit's orientation selectivity, suggests that while some units outside of the highlighted magnocellular population that are tuned to higher frequencies have a tendency to be indeed more orientation-selective, this pattern remains relatively weak.

In summary, training with the biomimetic regimen results in the emergence of a relatively homogeneous magnocellular group of units, which is markedly absent in the standard network, as well as receptive field types that are more aligned with parvocellular characteristics. These general patterns also hold for the other random initializations of the network (Supplementary Fig. 2), the other four network settings (Supplementary

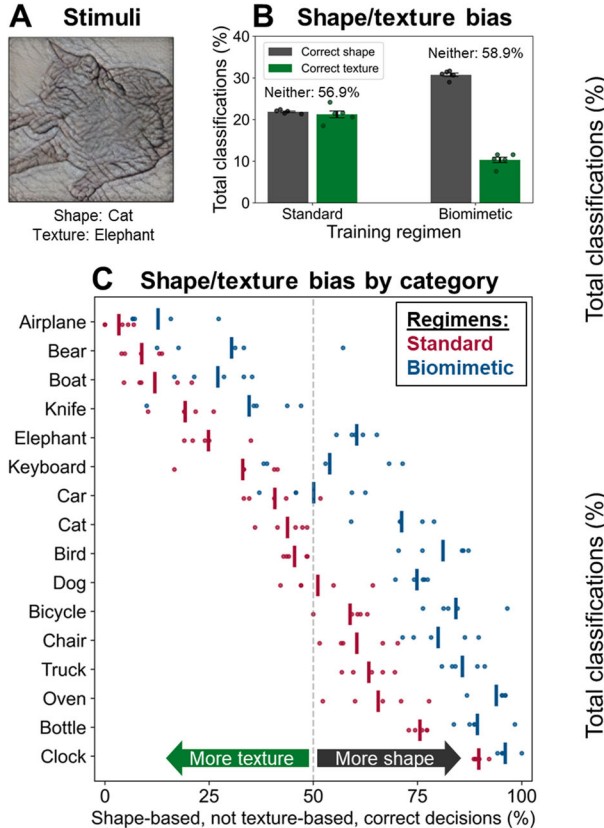

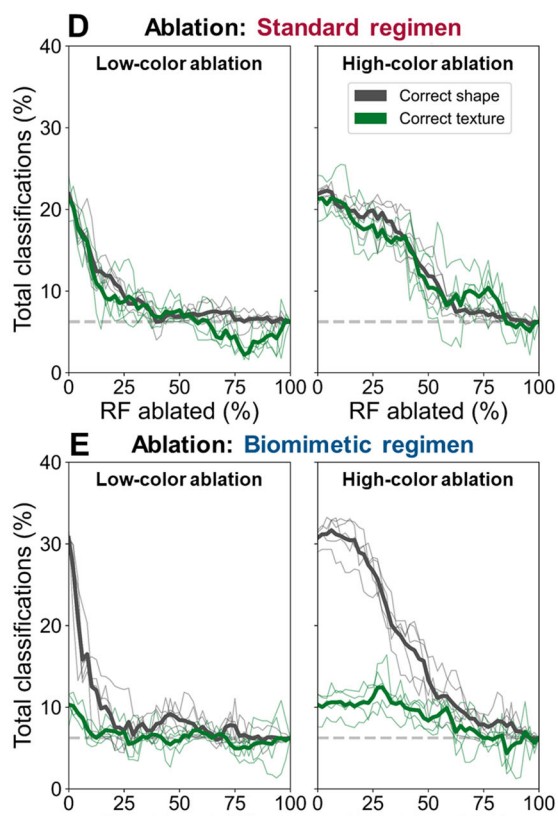

**Fig. 2 | Results of texture/shape analysis. A** Exemplar shape/texture cue conflict stimulus, reprinted from Geirhos et al.[45], depicting the local texture of an elephant and the global shape of a cat. **B** Percentage of total classifications correct in terms of shape, correct in terms of texture, or neither (i.e., incorrect). Error bars represent the standard error across the five different training runs with random initializations, and dots depict results of each individual run. **C** Percentage of shape-based correct classifications, as opposed to texture-based correct classifications, for each of the 16 different super-classes. Classifications that were inconsistent with both texture and shape classes are not included in this computation. Shown here are results of the five individual training runs (dots), along with their means (bars). Depicted on the y-axis are shape categories. Shape/texture bias as a function of the proportion of the ablated least color-sensitive (left) and most color-sensitive (right) units, for the standard (**D**) and biomimetic (**E**) regimen. Depicted are the five training runs (thin lines) as well as their means (thick lines). The dashed line, plotted at 1/16, indicates the chance classification level expected of a network generating random responses.

Figs. 3-10), and the three other biomimetic regimens tested (Supplementary Figs. 11-12).

## Texture/shape bias in classification decisions

Next, as motivated in the introduction, considering that the magno pathway is believed to be involved in more coarse-grained processing, while the parvo pathway is specialized in fine-grained spatial analysis, we examined the relationship between learned receptive fields and network behavior. An important dimension in this regard is that of local texture versus global shape encoding. To systematically probe it, we tested whether classification decisions are biased toward texture or shape, using the texture-shape conflict methodology previously established[45] (see Methods for details). Figure 2A depicts an exemplar image exhibiting a cue conflict between the local texture of an elephant and the global shape of a cat. As can be seen in Fig. 2B, C, the biomimetic model exhibits a markedly stronger bias to classify images based on global shape, indicating that its classification strategy is more similar to that of humans[45]. In contrast, the standard training did not lead to such bias. While there are some differences across the 16 different classes, the results of both models are fairly consistent across classes. Similar results also hold for all other settings and biomimetic regimens tested (Supplementary Figs. 13–17).

Further, we sought to determine whether the population of units exhibiting magnocellular characteristics might indeed enable the stronger shape bias of the biomimetic model. To this end, we gradually eliminated (or 'ablated') the most color-tuned (i.e., more parvo-like) vs. the least color-tuned (i.e., more magno-like) first-layer receptive fields of the trained

networks and re-computed the shape bias. The ablation of each receptive field involves ablation of all its (here, 22 × 22 × 3) values. This analysis, depicted in Fig. 2D, E, reveals that ablating less than 25% of the least color-sensitive receptive fields eliminates the shape bias of the biomimetically trained network entirely. By contrast, eliminating the same proportion of the most color-tuned receptive fields does not have a comparable impact. For the 'standard' network, we observe a less differentiated effect. Repeating the same experiment by ablating the receptive fields most tuned to high frequencies (i.e., more parvo-like) vs. low frequencies (i.e., more magno-like) reveals qualitatively similar results, albeit marginally weaker (Supplementary Fig. 18). Similar results hold for all other settings and biomimetic regimens tested (Supplementary Figs. 13–17).

In summary, training with our biomimetic regimen induced not only more human-like classification decisions based on global shape; it also revealed the potential role of receptive fields exhibiting magnocellular characteristics in supporting such global shape bias.

## Unit ablation and invariance studies

To complement results on shape versus texture bias in encoding, we also examined the effects of ablation on the classification performance of both models (Fig. 3A, B). Similar to the differences reported in Fig. 2D, E, classification performance of the biomimetic model is more differentially affected than the standard one by the ablation of low color or low spatial frequency units, relative to the ablation of high color or high spatial frequency units. This is evident by comparison of the left vs. right panels of Fig. 3A, B.

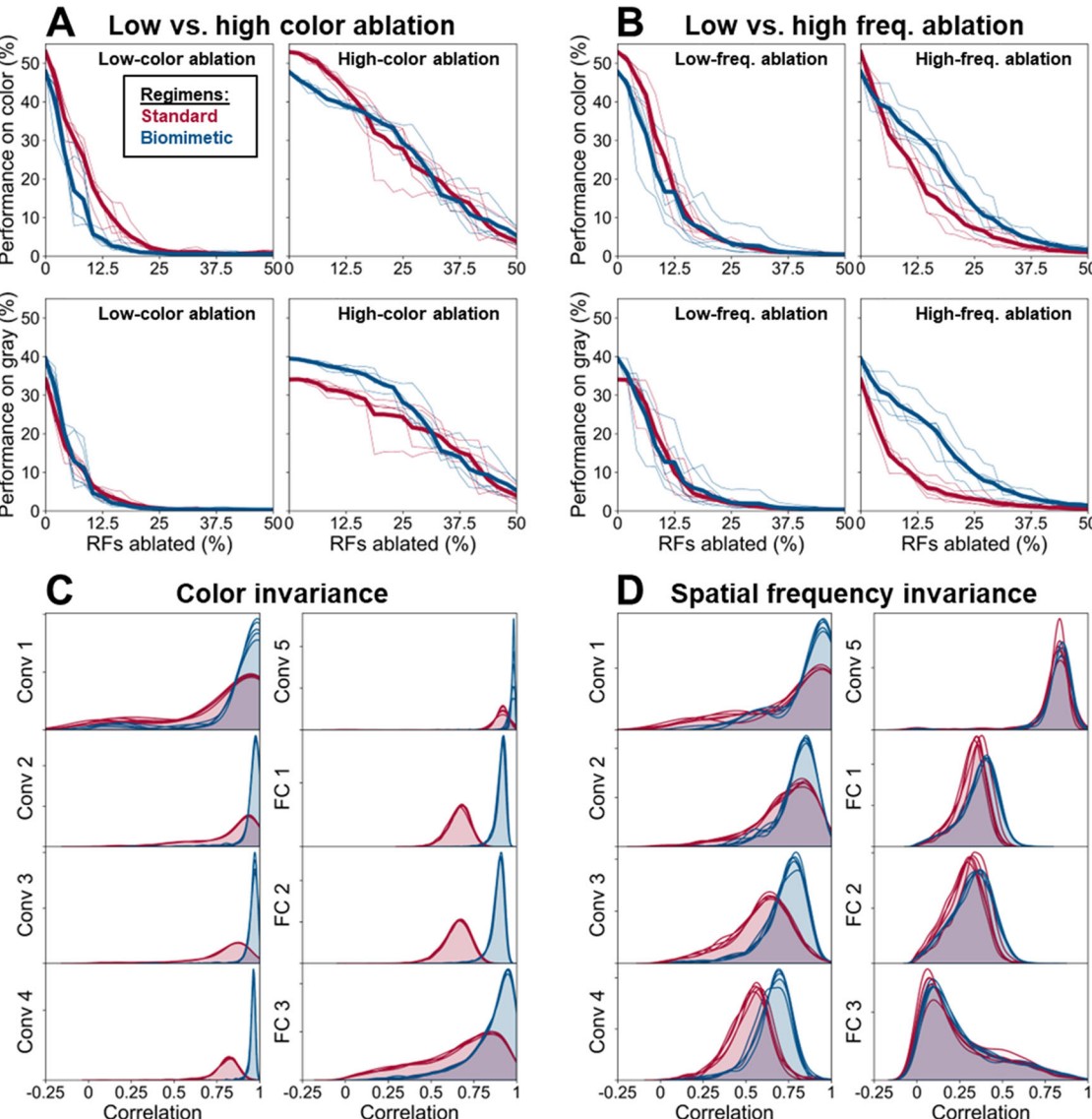

**Fig. 3 | Additional characterization of training regimens.** Classification performance on color (top) and grayscale (bottom) images when ablating the least vs. most color-sensitive (**A**) as well as least vs. most high spatial frequency tuned (**B**) first-layer receptive fields. Depicted are results of five training runs with different random initializations (thin lines), along with their means (thick lines). As there are 1000 ImageNet classes, the chance level is 0.1%. Distribution of correlations of neural activations across layers between full-color and grayscale images (**C**) and between full-frequency and blurred images (**D**). Depicted are superimposed results of the five individual network training runs.

Further, performance curves remain more similar across full-color vs. grayscale images for the biomimetic network than for the standard one. This effect may be due to increased invariance of the biomimetic model to the removal of chromatic information. To quantify the invariance (i.e., maintained similarity of activations) of both networks to the removal of chromatic as well as high spatial frequency content, we computed the distribution of correlation coefficients across all units for activations elicited by color vs. grayscale images (Fig. 3C) and full-frequency vs. blurred images (Fig. 3D) (see Methods for details). For both stimulus dimensions, the obtained distributions lean towards higher correlation values for the biomimetic than for the standard model, suggesting greater invariance of the biomimetic model. Notably, this is the case across all network layers. Thus, while we previously focused on the first convolutional layer for analyses of the joint coding of spatial frequency and color sensitivity in individual receptive fields, some of the biases in terms of color and spatial frequency encoding appear to be maintained in deeper network layers as well.

## Temporal characteristics of magnocellular-like units

The investigations above revealed that biomimetic training leads to the emergence of a group of receptive fields that exhibit magnocellular-like characteristics in terms of chromatic and spatial frequency tuning as well as their functional roles in classification behavior. Another key dimension that distinguishes magno and parvo neuronal sensitivities is the temporal one. Specifically, magno cells have been reported to have higher temporal frequency sensitivity[8–10], allowing the magno stream to encode more rapid temporal transients. Extending our investigation to the temporal domain, we incorporated a time dimension into our deep convolutional neural network (thereby making it a '3D'-CNN) and trained it on an action recognition video database[46] using both a spatially-biomimetic (i.e., low-to-high color and spatial frequency input) and a standard regimen (see Methods for details).

First, this investigation revealed that similar to the results obtained with 2D-CNNs, spatially-biomimetic training with 3D-CNNs also leads to 3D receptive fields exhibiting lower color and lower spatial frequency tuning

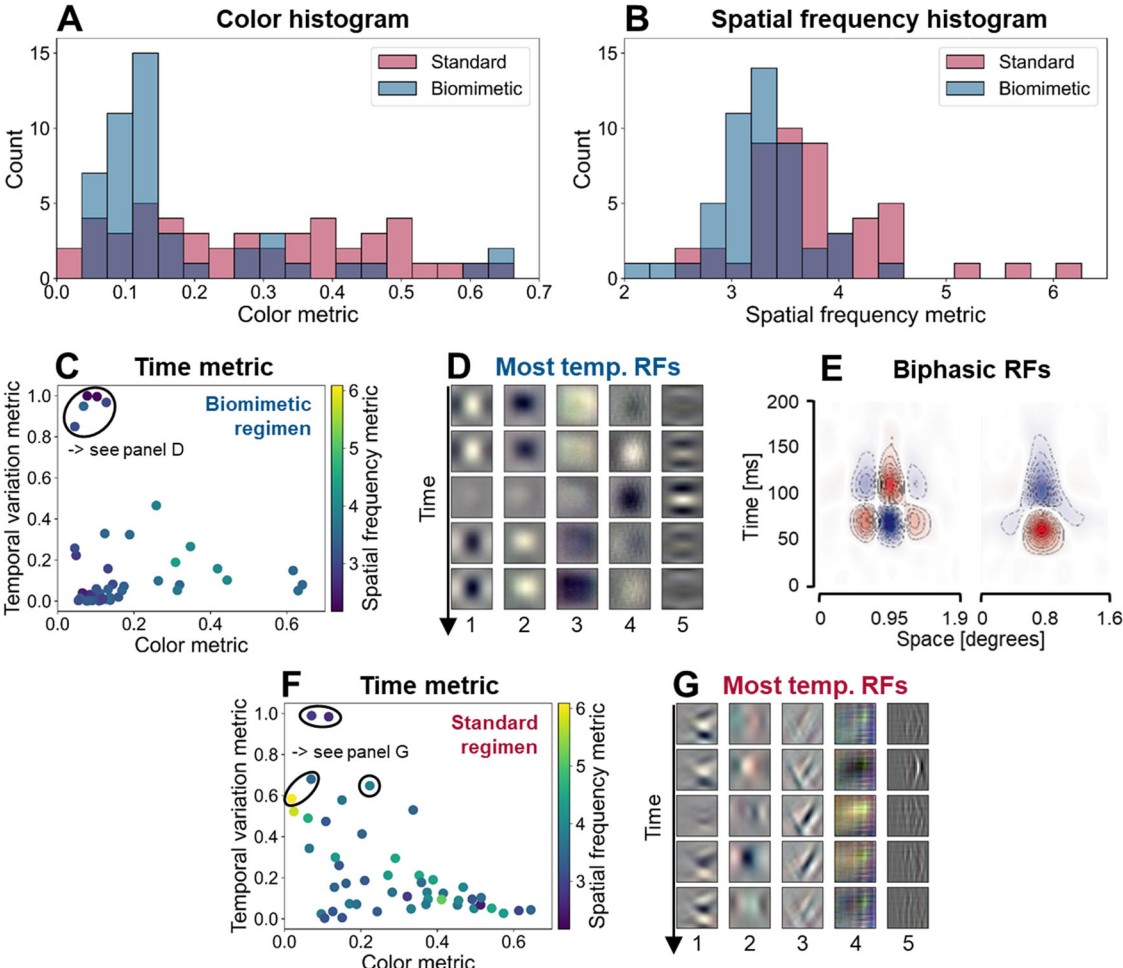

**Fig. 4 | Results of simulations with 3D-CNNs.** Histograms of color (**A**) and spatial frequency (**B**) metrics for receptive fields of the biomimetic and standard model when training is conducted with video inputs to a 3D-CNN. **C** Relationship between temporal RF properties (using the temporal variation metric; see Methods) and spatial RF characteristics (using the color and spatial frequency metrics akin to those used before) of the biomimetic model. The ellipse marks the five receptive fields exhibiting the greatest temporal variation. These RFs also show consistently low color and spatial frequency tuning. **D** Visualization of the five most temporally varied receptive fields of the biomimetic model. **E** Depiction of biphasic receptive field, adapted from the neurophysiological study by De Valois et al.[12]. **F** Relationship between the temporal variation metric and spatial RF characteristics (color and spatial frequency metrics) of the standard model. The ellipses mark the five receptive fields exhibiting the greatest temporal variation. **G** Visualization of the five most temporally varied receptive fields of the standard model. All plots in Fig. 4 have been generated based on the first out of five training runs with different random initializations. Results of all five training runs are shown in Supplementary Fig. 19.

(Fig. 4A, B). This finding emerges consistently across the five training runs with different random initializations (Supplementary Fig. 19).

We next examined the relationship between these two spatial properties and temporal receptive field dynamics, employing a metric capturing the maximum anti-correlation across time frames of a given 3D receptive field (see Methods for details). This analysis reveals that units exhibiting maximal temporal variation in the biomimetic model (i.e., units aligned with magnocellular properties in the temporal domain) indeed show fairly consistent magnocellular characteristics in the spatial domain (i.e., low chromatic and spatial frequency tuning) (Fig. 4C; Supplementary Fig. 19 for other training runs). Such units, depicted in Fig. 4D, exhibit strong similarity to biphasic cells (illustrated in Fig. 4E) predominantly found in the magno pathway[12]. It is important to note, however, that most receptive fields generally exhibited relatively low temporal variation (possibly due to the task requiring only moderate incorporation of temporal cues for classification success). Thus, while the most temporally varied receptive fields in the biomimetic model exhibit spatially-magnocellular response properties, there are several spatially-magnocellular units that do not exhibit strong temporal variation. The receptive fields exhibiting the least temporal variation are depicted in Supplementary Fig. 20. Interestingly, for a 3D-CNN trained with the standard regimen, several of the most temporally varied receptive fields tend to also be associated with relatively low-frequency, achromatic tuning (Fig. 4F; Supplementary Fig. 19 for other training runs). However, this association is less clear-cut than for the biomimetic regimen, with some maximally temporal receptive fields also tuned to higher spatial frequency or color content (Fig. 4G).

In summary, receptive fields exhibiting magnocellular characteristics in the spatial domain (low chromatic and spatial frequency tuning) also appear to exhibit properties that are in agreement with neurophysiological observations in the temporal domain (high temporal variation, consistent with higher temporal frequency sensitivity). This relationship holds for the biomimetic model and, to a lesser extent, for the standard model.

## Discussion

In this paper, we have presented a developmentally inspired account of the emergence of some of the differential response properties of units in the parvo- and magnocellular pathways, along with computational tests of its predictions. Our findings suggest that the joint coding of low spatial frequency and low color information in some receptive fields, and high spatial frequency and/or high color sensitivity in others, might be an outcome of the co-occurrence of these properties at different developmental time points. Training with the biomimetic regimen resulted in a magnocellular-like

population of receptive fields exhibiting both low spatial frequency and low color tuning. Biomimetic training also led to parvocellular-like receptive fields. However, rather than all being tuned to high spatial frequencies and color content, these units exhibited greater heterogeneity in terms of their tuning profiles. As described in the introduction, in light of the two sub-populations of parvocellular units (blobs and interblobs) observed at the level of the primary visual cortex[7], it is interesting to note the analogous heterogeneity among parvocellular-like units in the simulations.

In the temporal domain, we found that receptive fields exhibiting magnocellular-like properties in terms of spatial attributes (i.e., having low spatial frequency and color tuning) also tend to show higher variation across the time axis. This computational outcome aligns qualitatively with neurophysiological observations that magnocellular neurons respond to higher temporal frequencies[8–10]. It is worth noting that a 3D-CNN trained with the standard regimen yielded a similar pattern of results, although to a lesser extent. This points to an interesting difference between the spatial and temporal dimension. Specifically, in the spatial domain, biomimetic training was necessary to produce a differentiation between magnocellular-like (low color, low spatial frequency) and parvocellular-like (higher color, higher spatial frequency) units. In contrast, an association between low spatial and high temporal frequency tuning and, thus, a specialization to either fine-grained spatial features without much temporal variation or coarse-grained spatial features with stronger temporal dynamics appears present, to some extent, even when following the non-biomimetic training scheme.

On the functional level, biomimetic training not only induced more human-like classification decisions based on global shape rather than local texture information; it also suggests that units with magnocellular-like properties may support this global shape bias. This finding is in agreement with the magnocellular system's involvement in global organization[1,13,15–17]. Our study also adds to recent work reporting an increased shape bias when networks were trained with a mix of blurred inputs (including strongly blurred ones)[30]. Specifically, given the proposed linkage between receptive field properties and global shape biases, we present a hypothesis about how such shape biases might arise at the representational level. Moreover, we demonstrate that increased shape biases can also emerge in developmental settings (that is, for progressions transitioning from poor to rich), and we have also incorporated joint trajectories in both color and spatial frequency.

In addition to the biomimetic model exhibiting more magnocellular-like receptive fields and a stronger shape bias, its classification performance was also more differentially affected than that of the standard network by the ablation of low color or low spatial frequency units than by the ablation of high color or high spatial frequency units. The biomimetic model also exhibited stronger invariance in terms of its neural activations to the removal of chromatic or high spatial frequency content. In interpreting comparative benefits of the biomimetic vs. standard training regimens, it is important to acknowledge, however, that biomimetic training requires more epochs to achieve stable performance levels, as indicated by slower convergence of training loss (Supplementary Fig. 21).

Our computational results obtained with the non-biomimetic regimen suggest that dispensing with initially degraded inputs characteristic of normal visual development reduces global shape processing as well as the presence of magnocellular-like units tuned to both low spatial frequencies and colors. Interestingly, this result is in part corroborated by past behavioral and neural data.

Behavioral evidence derives primarily from studies of individuals who were born blind and gained sight late in life through cataract surgery[33]. Notably, these individuals differ from normally-sighted controls not only in their pre-surgical deprivation but also in terms of the quality of their visual experience post-surgery. As many of the maturational processes, which are significantly driven by retinal development[47], continue despite the children's blindness, their post-surgical visual system is markedly more mature than that of a newborn[47–50]. Indeed, their behaviorally assessed visual acuity, while below that of typically developed controls, is above neonate levels[51], and their color sensitivity reaches normal levels even just a few days post-surgery[32]. In other words, the late-sighted commence visual experience with

fairly high-quality inputs right from the start of their post-operative visual journey and thereby skip much of the initially degraded period that is characteristic of normal visual development. Our simulations predict detrimental representational and functional consequences of such a start with abnormally high-quality input. Consistent with this prediction, late-sighted children were found to exhibit deficits in tasks requiring extended spatial processing, such as face recognition and configural face judgements[28,52–54]. Similarly, the late-sighted were reported to exhibit deficits in global motion processing[55] as well as global shape completion and illusory contour perception[56]. Relative to resolution-acuity-matched controls, late-sighted children also exhibit reduced vernier acuity[57]. As an instance of hyperacuity, vernier offset detection has been attributed to extended spatial organization and multi-unit cortical processing[58–60]. Finally, late-sighted individuals were found to show deficits in object recognition generalization to color-removed images[32]. While we cannot rule out the possibility that some of these deficits could arise from deprivation during critical periods specific to certain visual functions[52], the reduced access to early degraded inputs provides a more parsimonious, unifying account.

In addition to these behavioral linkages, neural evidence derives from reversible suture experiments in monkeys. Specifically, artificially induced early deprivation followed by later restoration of sight was found to lead to greater long-term damage to the magnocellular than to the parvocellular units in the visual cortex[61]. In light also of the magnocellular system's stronger inputs and greater maturity at birth[62–64], its greater susceptibility to early deprivation has been accounted for by the earlier onset of its effective inputs. While these magnocellular deficits could, in principle, be accounted for by an anatomically hard-wired and pathway-specific critical period, following which, due to reduced plasticity, sight restoration does not allow the magnocellular pathway to gain normal function, an alternative account may be based on the lack of initially degraded inputs, preventing the magnocellular pathway from developing normally. Our computational simulations add plausibility to this proposal, considering that training with high-quality visual inputs from the beginning, as opposed to initial training on low-frequency, achromatic stimuli, resulted in the marked absence of receptive fields exhibiting magnocellular characteristics.

Although the above findings support the idea that the magnocellular pathway may be especially reliant on initially degraded visual input for its normal development, some experimental observations are not entirely consistent with this account. As reviewed earlier, the magnocellular pathway is associated with higher temporal frequency responses. Thus, one might expect late-sighted individuals to exhibit deficits in temporal sensitivity commensurate with their spatial deficits. However, empirical studies indicate that temporal vision is more resilient than spatial vision to early-onset visual deprivation[65,66]. Even in the spatial domain, although the late-sighted surpass neonates in terms of visual acuity, they still fall short of typically developed adults[51]. These findings suggest a more nuanced picture, in which certain aspects of magnocellular function may be less susceptible to atypical early experience and in which parvocellular functions are not entirely spared under such conditions.

Several additional considerations and potential limitations merit discussion. The first is inherent in any computational study that relies on the use of deep neural networks. Despite the successes of deep networks in modeling certain aspects of visual processing, they still diverge from human vision in important ways. Significant discrepancies have been highlighted between deep network behavior and findings from human psychology[36]. Indeed, many studies have shown that DNNs differ from humans in such aspects as global shape encoding[67,68], Gestalt grouping[69], crowding[70,71], as well as generalization and robustness[72]. As Wichmann & Geirhos (2023) describe, deep networks may be "promising" but not yet fully "adequate" models of visual processing[35]. However, pinpointing mismatches between the biological and artificial systems can lead to insights into the former and iterative refinements of the latter[41,73]. In the context of the present study, introducing developmentally inspired input progressions offers potential insight into the emergence of aspects of the biological system and illustrates

one way in which known limitations of deep networks in spatially global processing could be mitigated.

A second limitation concerns architectural details. Our model does not incorporate lateral and/or top-down recurrence, which has been proposed to be important for challenging tasks involving occlusion or noise[74,75], for capturing the representational dynamics of the human ventral stream[76], and for supporting complex analyses beyond the initial feedforward sweep[77]. Moreover, parvocellular neurons are known to vastly outnumber magnocellular ones in the fovea, and the cortical representation of the fovea is disproportionately large[78]. Various architectural constraints, such as cortical magnification, convergent feedforward projections, and interhemispheric connections have also been shown to critically affect the interpretation of neural data[79]. In the current study, we chose a simplified approach that does not incorporate these constraints, in part because such constraints are known to change throughout development. For instance, feedback connections are reported to emerge a few months after feedforward ones[80,81], rendering the architecture of the initial visual experience more feedforward. Similarly, the fovea is notably more immature at birth than the parafovea[82,83], which, too, may significantly affect neural encoding. Given these complex developmental factors, any architectural constraints incorporated may need to undergo changes throughout training, which presents its own challenges. While these important considerations lie beyond the scope of the present paper, we acknowledge that they could influence developmental outcomes and warrant detailed follow-up investigations.

A third limitation worth noting is that while our analyses focus on color sensitivity, spatial frequency, and temporal response characteristics, our learned receptive fields differ from those in real biological systems in several respects. For example, biological receptive fields often exhibit specific types of color opponency, whereas we adopt a simpler 'more vs. less' color metric. Moreover, while biomimetic training yields a population of magnocellular-like units and more heterogeneous parvocellular-like units, these distributions are not categorical and may represent a broader continuum. Finally, our analyses do not explicitly consider koniocellular (K) cells, which constitute a highly heterogeneous collection in the primate LGN and feature multiple subclasses of units with distinct response properties[84,85]. As Hendry & Reid (2000) note, the existence of this additional and highly heterogenous pathway "introduces a complexity into a field only recently viewed as clean and simple"[85]. Thus, while our approach captures some aspects of the organization in the mammalian visual system, a more comprehensive account will need to incorporate this additional complexity as well.

A fourth consideration relates to the complex interplay of 'nature' and 'nurture'. Our results show that, in principle, a generic visual system, when exposed to a biomimetic training progression, can develop a rough division of magnocellular-like and parvocellular-like units without the need for hard-coded separate pathways ab initio. In other words, our results demonstrate the possibility of an experience-driven ('nurture') route to such differentiation. At the same time, this demonstration does not rule out the possibility that, in the real biological system, some aspects of parvocellular and magnocellular organization already exist through phylogenetic dispensation ('nature') prior to patterned sensory inputs[63,64] and may be refined by subsequent experience. Attesting to this complex interaction, although some structures of cortical maps can form without actual sensory experience, the emergence of specific features of such maps has been reported to be critically experience-dependent[86]. It is also possible that both ontogenetic development (i.e., development within an individual's lifetime, driven directly by sensory experience) and phylogenetic development (i.e., the evolutionary shaping of neural architecture and biases across generations, possibly driven over the long term by statistics of the sensory environment) play complementary roles in establishing and refining the parvo- and magnocellular pathways.

Despite some of these open questions, our findings build on previous computational studies demonstrating that certain aspects of the biological visual system can be reproduced without reliance on any explicit innate biases. For instance, already earlier work demonstrated that receptive fields exhibiting some resemblance to those of the mammalian primary visual cortex can emerge through sparse coding on natural images[87]. More recently, deep learning models without domain-specific, innate biases were shown to develop sophisticated internal models of the world when exposed to child-like inputs[88]. Similarly, newborn chicks and generic deep networks, when exposed to the same impoverished environments, have been shown to learn similar object recognition abilities – supporting an experience-driven account of development[89]. Collectively, these studies illustrate the power of computational modeling in informing longstanding nature-nurture debates[90]. While not ruling out innate constraints, they underscore how experiential factors can potentially give rise to representational organization, with developmental, computational, and evolutionary implications.

Finally, the finding that training with a developmentally-inspired progression of inputs yields representations more consistent with empirical neurophysiological results, and induces more human-like global shape-based processing, adds to a growing body of literature demonstrating the benefits of 'adaptive initial degradations' across different sensory domains, such as visual acuity[28], color vision[32] and prenatal hearing[91,92]. The principle emerging from these studies is that initially impoverished inputs may be adaptive and provide a scaffold rather than act as hurdles for the acquisition of later perceptual skills[33,93–95].

In conclusion, the work presented here offers a potential account for the emergence of the parvo/magno distinction and thereby also provides a framework for understanding the potential adaptive significance of why normal development progresses in a particular temporal sequence. It further helps account for some of the impairments associated with atypical perceptual development and, on an applied note, demonstrates how findings from biological development can help develop useful training protocols for computational systems[96].

## Methods
### Convolutional neural network model (for 2D simulations)
For the simulations reported in this paper, we used the AlexNet[43] and carried out training and testing across the following different settings:

- For setting 1 reported in the main manuscript, we slightly adapted the original architecture. First, we enlarged the size of the RFs in the first convolutional layer from $11 \times 11$ to $22 \times 22$ pixels to avoid restricting the RF structures that can be learned by the network, and to allow for the extraction of frequency-based metrics with higher resolution and precision. Second, the number of first-layer RFs was reduced from 96 to 48 as a precautionary measure to increase the likelihood of all RFs learning clear structures that can be reliably quantified using metrics. (Although originating from a different domain, in the context of learning genomic sequence motifs, past work has shown that too many first-layer convolutional filters can lead to more filters not learning meaningful representations[97]). Training lasted for a total of 200 epochs with a constant learning rate. These settings were kept deliberately simple (rather than featuring, for instance, complex decreasing learning rate schedules) in order to render the number of epochs more interpretable, facilitate simple replicability, and ensure that any potentially beneficial effects emerging from the first half of training in the biomimetic model are not just due to reduced learning rates in the second half. To ensure fair and stable convergence of all models, we employed a relatively extended number of epochs. It is worth noting that biomimetic models require more epochs to reach stable levels of loss and accuracy, as illustrated in Supplementary Fig. 21. With this, we were also able to obtain well-established RF structures that can be analyzed meaningfully and reliably. However, to examine the generalizability of our findings, we systematically varied several of these parameters, as described below.
- For setting 2 reported in the supplementary material, we built on setting 1 but used the full set of 96 first-layer RFs (when enlarged to $22 \times 22$ pixels).
- For setting 3, we built on setting 1 but used the original number and size of RFs in the first layer (i.e., a total of 96 RFs, each of size $11 \times 11$ pixels). This setting was included to verify whether even with the original architecture, the basic results would hold.
- For setting 4, we used setting 1 but only trained for a total of 100, instead

of 200, epochs. This setting was included to ensure that the observed findings are not just the consequence of a comparatively long total training duration.

– Finally, for setting 5, we used setting 1 but, to examine generalizability to more modern training procedures, implemented a decreasing learning rate schedule (for details, see below).

## Network training

The (slightly adjusted) AlexNet was implemented and trained using Keras / TensorFlow v2. We utilized the official split of the ImageNet database[44] into a training set (containing more than 1 million images belonging to 1000 different object classes) and a test set (containing a total of 50,000 images – 50 for each object class). For training, we used the categorical cross-entropy as loss function and Stochastic Gradient Descent (SGD) as optimizer, with a Nesterov momentum of 0.9 and a batch size of 128. For settings 1–4, we chose a constant learning rate of 0.001. For setting 5, we used a decreasing learning rate schedule, monitoring the validation loss, with an initial learning rate of 0.02, reduction factor of 0.5, patience of 10, minimum delta of 0.0001, cooldown of 0, and a minimum learning rate of 0.0001. The train/validation loss and accuracy throughout training are visualized in Supplementary Fig. 21. For all settings, image preprocessing and augmentation were kept fairly minimalistic: random $227 \times 227$ segments were cropped out of the full $256 \times 256$ images, pixel values were rescaled from a $[0,255]$ to a $[-1, 1]$ distribution, and images were flipped horizontally at random. Blurring (for the developmentally inspired training regimens as well as for testing in Fig. 3) was accomplished by applying a Gaussian blur with what would correspond to a sigma of 4.

## Different training regimens

The above settings (1–5) are used to train on both the 'standard' and 'biomimetic' training regimen:

– In the 'standard' regimen, training lasted for a total of 200 epochs in all settings except setting 4 (100 epochs in setting 4) and contained exclusively high-resolution, full-color images.

– In the 'biomimetic' regimen, training on blurred, grayscale images for 100 epochs (for 50 epochs in setting 4) was followed by training on high-resolution, full-color images for another 100 epochs (for 50 epochs in setting 4).In addition, the following variations on biomimetic regimens were trained using setting 1:

– In the 'biomimetic v2' regimen, training on low-resolution, grayscale images for 50 epochs was followed by training on low-resolution, full-color images for 50 epochs, and training on full-color, high-resolution images for 100 epochs. This was inspired by the faster development of color than acuity, resulting in an intermediate stage where color sensitivity is fully developed but visual acuity is not.

– In the 'biomimetic v3' regimen, training on blurred, grayscale images for 100 epochs was followed by training on high-resolution, full-color images for 200 epochs. This regimen was included in light of the relatively short duration of initially degraded visual experience in human development.

– In the 'biomimetic v4' regimen, training on blurred, grayscale images for 100 epochs was followed by training on high-resolution, full-color images for 100 epochs (as in the 'biomimetic' regimen), but a proportion (here, 50%) of the first-layer filters was confined to only learn during the second half of training. This serves as an additional biomimetic control to account for the possibility that parvocellular units may effectively develop after magnocellular ones and may thus not have been exposed to maximally degraded stimuli.

## Spatial frequency RF metric

To measure spatial frequency content, we applied a 2D-FFT to a grayscale version of each RF and used radial averaging to summarize the presence of different spatial frequencies, providing us with a 1D vector as a function of frequency. Given such vector, we defined our spatial frequency metric as the weighted average frequency:

$$weighted\ average\ frequency = \frac{\sum_f amp(f) * f}{\sum_f amp(f)}$$

Amp thereby refers to the amplitude of a given frequency, and f refers to the frequency itself. There are a few details worth pointing out. First, the constant part of the FFT was excluded for the calculation. Further, in order to avoid any noise that may be caused by the discreteness of the index, all RFs were up-sampled by a factor of 100 prior to the application of the 2D-FFT. Note that our metric, due to its denominator dividing by the total sum of amplitudes, is independent of the absolute strength of the signal, rendering the question of whether to use a normalized or unnormalized spectral decomposition obsolete.

Finally, note that in neurophysiological studies, a common technique to summarize spatial frequency tuning curves is to simply extract the frequency eliciting maximal activity[10,98]. Here, we chose to use the weighted average frequency metric to not just describe the frequency corresponding to maximal amplitude but to also take into account the amplitudes of all other frequencies. This is especially important given the rather small size of RFs (even if increased to $22 \times 22$ and up-sampled). Overall, for the specific characterization carried out here, this approach yields a more sensitive and fine-grained differentiation between individual RFs and prevents clustering around fairly discrete values.

**Orientation selectivity RF metric**. We also defined a metric to capture the extent of orientation tuning exhibited by individual RFs. Akin to the computation of the spatial frequency metric, we started by carrying out a 2D-FFT on each RF. As opposed to radial averaging, where direction-independent frequency profiles are extracted, we here applied azimuthal averaging, where frequency bands are averaged across. Note that the utilized frequencies were restricted to only a quarter of the theoretically available frequency spectrum – a compromise chosen to filter out some high-frequency noise while not removing any effective frequencies in the RFs. Overall, this allows us to assess intensity as a function of orientation, ranging from zero to π. Orientation-tuned RFs are associated with a sharp and strong peak in this distribution. This can be captured well by the mean resultant length[99]:

$$R = \frac{1}{\sum f(\theta)} \left| \sum_\theta f(\theta) e^{2i\theta} \right|$$

where θ represents the orientations from 0 to π, and f (θ) their corresponding azimuthally-averaged amplitudes. A line plot depicting the intensity over orientation from 0 to π can thereby be imagined to be plotted around a whole circle, with the origin of the 2D coordinate system in the center and orientations 90 degrees apart from each other on opposing ends of the circle. Our metric then represents the length of the vector resulting from averaging in this coordinate system. This computation is inspired by the 'circular variance' metric widely used in neurophysiological studies, where, similarly, the resultant is applied to 1D tuning curves as a function of orientation to extract how orientation-selective a given cell is[100].

## Color RF metric

Physiological studies have often characterized neurons in terms of what specific colors or color opponency they are most sensitive to. Of greater relevance here, other measures have been established in neurophysiology[101], which were subsequently used in computational modeling studies[102], that characterize the overall sensitivity/responsivity to color based on the strength of responses to color vs. grayscale stimuli. Here, working with learned filters rather than recorded responses, and allowing for the analysis of colorfulness of sub-sections of the RF (to not introduce biases in estimating spatially-extended RFs as more colorful), we quantified the color

sensitivity (or colorfulness) of a given RF in two steps. First, we extracted the discrepancy of individual color channels, $m$, across the R, G, and B channels for each individual pixel:

$$x = R\cos 0° + G\cos 120° + B\cos -120°$$

$$y = R\sin 0° + G\sin 120° + B\sin -120°$$

$$m = \sqrt{x^2 + y^2},$$

where R, G, and B represent the pixel-by-pixel channel intensities. We then summarized the distribution of such color channel discrepancies across the $22 \times 22$ pixels of a given RF into a single value. Defining our final color metric as the mean of the $22 \times 22$ distribution would induce an estimation bias towards spatially extended color RFs. Instead, taking into account only the single most colorful pixel could potentially be subject to high noise. As a compromise, we chose to define our final metric as the average of the top-48 (for settings 1, 2, 4, and 5; the top-12 for setting 3; i.e., approximately the top-10%) most colorful pixels within a given RF, roughly approximating the size of the smallest effective RFs. Note, however, that qualitatively similar results hold when taking even fewer of the most colorful pixels into account.

While we have chosen the RGB color space to characterize deviations from grayscale (by quantifying imbalances between R, G, and B values), one could have done so in other color spaces as well. Supplementary Fig. 22 shows the results (for setting 1) when extracting the saturation (S) in HSV space or chroma ($\sqrt{a^2+b^2}$) in CIELAB color space instead. Overall, qualitatively similar results hold.

### Texture/shape bias analysis

For the results reported in Fig. 2 (and corresponding Supplementary Figs. 13–18), we present a texture/shape bias analysis using the previously introduced methodology[45]. The test images from their study are publicly available at: https://github.com/rgeirhos/texture-vs-shape/tree/master/stimuli/style-transfer-preprocessed-512. This dataset[45] contains a total of 1280 images in which global shape and local texture cues are in conflict. For instance, one image might exhibit the global shape of a cat but the local texture of an elephant. If a network classifies such an image as a cat, its decision would be deemed shape-consistent. If, on the other hand, the network labels such an image as an elephant, it would be judged as texture-consistent. Classifications that do not match either of these categories are considered incorrect (depicted as 'neither' in Fig. 2).

While ImageNet encompasses a total of 1000 image classes, the shape-texture methodology consolidates a subset of these into 16 broader, relatively high-level categories (listed on the y-axis of Fig. 2C), each representing a superset of the original ImageNet classes (the mapping can be found in the original repository: https://github.com/rgeirhos/texture-vs-shape/blob/master/code/helper/human_categories.py). The 1280 images were generated such that there are 5 images for each possible pairing of these 16 shape classes with the same 16 texture classes. Due to this design, 80 of the 1280 images include matching shape and texture cues. Following the analysis methodology used in the original paper, we excluded those 80 images from our shape-texture analysis[45].

For each of the relevant 1200 images, we obtained the network's final-layer predictions (1000 probabilities, one per ImageNet class). Because not all of the 1000 ImageNet classes map onto the 16 broader categories, we restricted our analysis to only those classes that do. We selected the class with the highest probability from this subset and mapped it to one of the corresponding 16 categories. Based on whether the chosen class matched the shape or texture of the shape-texture conflict image, the network's response was labeled shape-consistent, texture-consistent, or incorrect (if the class belonged to one of the remaining 14 categories). Note that in Fig. 2C, results are shown for each shape category separately. Specifically, for each of the 16 shape categories, we show the models' responses to the 75

images (1200 total images / 16 categories) in which the corresponding shape was present.

### Invariance analysis

In Fig. 3C, D, we present an invariance analysis aimed at evaluating how stable a network's unit activations are when input images are manipulated. Specifically, we computed distributions of correlation coefficients by comparing network activations produced by full-color, non-blurred images against activations produced by the same images when presented either in grayscale (Fig. 3C) or when blurred (Fig. 3D). Our goal was to examine whether the biomimetic model, compared to the standard network, maintains more similar responses under these alterations.

For computational efficiency, we began by subsampling the ImageNet test set. While the original test set comprises 50,000 images (50 images per ImageNet class), we randomly selected three images per class, producing a dataset of 3000 test images, which was used for all subsequent invariance analyses. Each of these 3000 images was then fed into both the biomimetic and standard networks in three forms: (1) as undegraded images (i.e., in full-color and non-blurred), (2) converted to grayscale, and (3) blurred with a Gaussian filter (sigma = 4).

For each of the networks, each of the 3000 test images, and each of its three variants (undegraded, blurred, and grayscale), we extracted the activations of every unit across all convolutional and fully-connected layers. We then concatenated these activations across the 3000 images for each of the three variants. As an example, consider the first convolutional layer of our minimally-adjusted AlexNet in setting 1. This layer has 48 filters, and for each filter, the activations corresponding to each of the three image variants can be represented by a $22 \times 22 \times 3 \times 3000$ matrix (height × width × number of color channels × number of test images). Each matrix was then flattened into a single 1D vector, such that every filter produced three vectors – one each for undegraded, grayscale, and blurred images.

Using these vectors, we computed correlations to quantify the similarity of responses across image alterations. For each unit (e.g., each filter in the first convolutional layer), we correlated its undegraded-image activation vectors with its grayscale-image activation vector and, separately, with its blurred-image activation vector. Figure 3C, D shows the distributions of these correlation values for all network layers. In the example of the first convolutional layer, each distribution comprises 48 correlation values – plotted in blue for the biomimetic model and in red for the standard model. Similar distributions were computed for all other convolutional and fully-connected layers.

### Convolutional neural network model (for 3D simulations)

We used the minimally adjusted AlexNet architecture from 'setting 1' (with 48 RFs of size $22 \times 22$ pixels) and extended it to incorporate a temporal dimension. The resulting architecture is described in Supplementary Table 1.

### Network training

We trained the above model on the Kinetics-600 action recognition database[46], which, at the time of downloading and after deletion of corrupted or too short videos, comprised 424,885 videos belonging to a total of 600 action categories. The original videos were temporally cropped to comprise the central 48 time frames. They were spatially resized to $128 \times 128$ pixels, based on the smaller of the height or width dimension and using a central crop of the longer dimension. Similar to the 2D-CNN, image preprocessing and augmentation were kept simple: random 32 (temporal) x 112 (spatial) x 112 (spatial) segments were cropped out of the 48 (temporal) x 128 (spatial) x 128 (spatial) videos, pixel values were rescaled from a [0 255] distribution to a [−1, 1] distribution, and videos were flipped horizontally at random. Blurring (for the biomimetic training regimen) was accomplished by applying a Gaussian blur with what would correspond to a sigma of 4. For training, we chose a batch size of 128, a

constant learning rate of 0.001, categorical cross-entropy as loss function, and Stochastic Gradient Descent (SGD) as optimizer, with a Nesterov momentum of 0.9. Similar to the training of our 2D-CNNs, we repeated training with five different random initializations each.

### Different training regimens

We trained the network using two different regimens:

– In the 'standard' regimen, training lasted for a total of 240 epochs and contained exclusively high-resolution, full-color videos.
– In the 'biomimetic' regimen, training on blurred, grayscale videos for 120 epochs was followed by training on high-resolution, full-color videos for 120 epochs.

### Color and spatial frequency metrics

For extracting the color and spatial frequency metrics of the 3D RFs, the 2D metrics (see above) were computed separately for each of the 5 time points of each RF, following which the average across the 5 time points was computed.

### Temporal metric

For each RF, we computed the (flattened, 1D) correlations between all pairwise comparisons of the 5 time points of a given RF. To extract the maximal temporal change across the entire temporal extent of a given RF, we computed the minimum of these correlation values. If the RF is identical across five time points, this value would correspond to +1. In case of a complete reversal of RFs (akin to 'biphasic receptive fields' illustrated in Fig. 4E), the value would be -1. As we wanted to determine the temporal variation, and not temporal stability, we mapped these minimum correlation scores to a temporal variation metric as follows:

$$temporal\ variation\ metric = \frac{1 - \min(correlation)}{2}$$

Thus, a minimum correlation of 1 would map onto a temporal variation metric value of 0, and a minimum correlation of -1 would map onto a temporal variation metric value of 1.

### Statistics and reproducibility

This is a purely computational study, and formal statistical tests were not performed. However, reproducibility was rigorously assessed by repeating all simulations across five independent runs, with means and standard errors reported. Additionally, parameter variations were tested to ensure generalizability of the results.

### Reporting summary

Further information on research design is available in the Nature Portfolio Reporting Summary linked to this article.

## Data availability

All data is available in our permanent Zenodo repository (https://doi.org/10.5281/zenodo.15543117)[103].

## Code availability

The code is available in our permanent Zenodo repository (https://doi.org/10.5281/zenodo.15543117)[103].

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

## Acknowledgements

This work has been supported by NIH grant R01EY020517 to Pawan Sinha. Lukas Vogelsang is supported by a grant from the Simons Foundation International to the Simons Center for the Social Brain at MIT. Marin Vogelsang is supported by the Japan Society for the Promotion of Science (JSPS), Overseas Research Fellowship and the Yamada Science Foundation.

## Author contributions

M.V., L.V., G.P., S.D. and P.S. conceptualized the study, M.V. carried out the computational simulations and analyses, M.V. and L.V. drafted the paper, and M.V., L.V., G.P., S.D. and P.S. wrote the final paper.

## Competing interests

The authors declare no competing interests.
