## [Transparent Peer Review file · Communications Biology]

Potential role of developmental experience in the emergence of the parvo-magno distinction

Corresponding Author: Dr Lukas Vogelsang

This manuscript has been previously submitted at another journal. This document only contains information relating to versions considered at Communications Biology.

Version 0:

Reviewer comments:

Reviewer #1

(Remarks to the Author)

This decade, DNNs have been increasingly explored as models of primate vision. Many such explorations use popular network architectures such as AlexNet (e.g., used in this study) and train these models on categorization tasks with massive image databases like ImageNet (as done in this study). Imposing biologically relevant constraints on these models is relatively rare at the level of their architecture, learning rules, as well as the “visual diet” or “regimen” used to train them. The specific simulation study reviewed here explores the consequences of manipulating the “training regimen” (or “visual diet”) on receptive field (RF) properties of network units. The study emphasizes that a “biomimetic” training regimen (first, consisting of low spatial frequency & achromatic images, next of standard chromatic images also including higher spatial frequency information) leads to network units revealing response properties that mimic those of neurons recorded from the mammalian visual cortex—i.e., magno- and parvo-cellular responses. The authors claim the observation of such rough correspondence among units of DNNs and those of cortical visual neurons supports their hypothesis that “[...] the temporal confluence in the progression of spatial frequency and chromatic sensitivities during development may significantly shape neuronal response properties characteristic of this division.” The authors further make two ancillary claims, that (i) “[...] biomimetic training induces a more human-like classification bias towards global shape” and (ii) their findings have “relevance for the design of training procedures for computational vision systems”.

In my view, the paper is well written and interesting in the sense that, like few previous studies (more on this below), it could help redirect attention in the field to the importance of biological constraints and network training regimens when it comes to interpret DNN performance measures and response properties of network units. I think the authors make a good case showing the importance of the statistical structure (both spatial and temporal) of images to achieve more human-like biases and physiological-like response properties of network units. In other words, their observations in my view support the second clause of the final sentence of their abstract in lines 23-25: “[...] and applied relevance for the design of training procedures for computational vision systems.” The reported analyses also support the claim that biomimetic training led to a human-like global-shape bias. However, I am not convinced the currently presented simulations and analyses convincingly support the first claim in that same sentence that: “These results have implications for understanding a key aspect of visual pathway organization”, or that the authors fully articulate an account of the emergence of the parvo- magno- distinction. I am therefore also not convinced the presented RF analyses provide “strong support” for their main hypothesis.

Although multiple aspects of the reported research are in my view creative and insightful (e.g., manipulating interpretable low-level aspects of the visual diet, as well as some ensuing RF analyses), and the figures well-crafted and informative, I am concerned the logic of the paper seems somehow circular, and the information in the methods section in my view insufficient to confidently evaluate the validity of the chosen metrics; especially given the lack of access to the code that implements them. The authors aim to explain the emergence of units in DNNs with magno- vs parvo-cellular properties by manipulating properties of the images (chromaticity and blurring) that do not seem to be independent from the RF-properties of those neurons whose emergence the authors are trying to explain.

The simulations explored also seem to ignore the fact that parvo-cells are highly concentrated at the fovea, and the primate ventral visual processing stream (believed to be engaged in shape representation and object recognition) exhibits a marked

foveal response bias. This is an issue the authors should in my view consider, as well as relevant related work.

In sum, although the paper is in my view interesting and contributes creative analyses and a fresh perspective that might positively influence the field, I feel that the logic underlying key arguments supporting their central hypothesis may require the examination of some main claims in the current version of the manuscript. Specifically, I am not convinced the authors provide an account of the emergence of the parvo- and magnocellular systems based on early sensory development, since it would require disentangling the influence of magno and parvo RF properties on the image transformations used to implement the “biomimetic” training regime—an admittedly very challenging task. The issues here are in my view reminiscent of the nature vs nurture debate. Both nature and nurture probably play a role, but unclear to me from reported simulations how to advance towards answering such questions. Below, I provide a more detailed list of major and minor concerns. If the authors addressed these concerns, this might in my view be an interesting and potentially impactful contribution.

Major concerns

1. Properly reviewing this paper would benefit from access to the code, images, and network models. In particular, it seems to me hard to evaluate the validity of some of the chosen metrics given the provided information (see comments to methods section, below).
2. Line 129: It is unclear for me why larger heterogeneity is observed for units tuned to high spatial frequency OR chromatic content. Also unclear if the OR is inclusive or exclusive and, whichever it is, why.
3. Line 143: It is unclear for me why the narrative of the paper shifts from M-P story to global shape & the Gheiros story. It is also unclear why the authors link global shape processing to magno-cells, and not parvo-cells. Further motivating these issues in the introduction as well as relevant references might help clarify this point.
4. Line 159: “we chose low color, rather than low spatial frequency as a proxy for M-units considering the larger homogeneity of the former, [...]”. This seems to me arbitrary. How do these analyses look if spatial frequency is chosen as proxy for M-units. Similar comment extends to line 165 regarding the causal role of RFs exhibiting magno characteristics.
5. Line 177: Unit ablation and invariance studies. Method to assess invariance incompletely clear, as well as how it may be influenced by possible normalizations (please, see comments to methods section further below)
6. Line 180-189 “[...] Further, the relative importance of these units remains more similar across full-color vs. grayscale images for the biomimetic network than for the standard network, presumably due to increased invariance to the removal of chromatic information. To quantify the invariance of both networks to the removal of chromatic and high spatial frequency content, [...] *This analysis reveals markedly greater invariance of the biomimetic model for both stimulus dimensions across nearly all network layers.*” (italics mine for emphasis)
7. Line 191: Analyses of invariance to the reduction of image contrast. It is not clear to me why the authors decided to explore contrast responses, or the validity of their analysis approach (see comments on methods section, below)
8. Lines 196-200: “This observation is aligned with the neurophysiological finding that magno cells have greater contrast sensitivity and thereby saturate with lower contrast (Shapley et al., 1981; Kaplan & Shapley, 1982; Hicks et al., 1983; Derrington & Lennie, 1984), *which would be expected to result in higher correlations between neural units across normal and low-contrast stimuli.*” (italics are mine for emphasis) This is to my understanding not necessarily true. Correlations need not change as a function of contrast even if responses revealed substantial contrast sensitivity. It seems also vague what the authors mean by “normal” and “low-contrast” stimuli. It would be helpful if the authors clarified the assumptions of their argument. A simple supplementary simulation may (or may not) help to clarify their point.
9. Line 213: Summary of analyses of temporal characteristics of magnocellular units. “The investigations above revealed that biomimetic training leads to the emergence of a group of receptive fields that exhibit consistent magnocellular-like characteristics in terms of chromatic and spatial frequency tuning as well as their functional roles in classification behavior.” Why do the authors believe they have probed and can relate the functional role of their magno-like units and real magno-cells? Is there evidence that real magno cells exhibit the functional properties described here? References and a more thorough discussion of this matter seems important.
10. Regarding the reported spatiotemporal RF analyses, it would be useful to compare results with similar analyses for the low temporal variation & low color units in the biomimetic and standard models. It would be also informative to report the distribution of RF sizes of network units, and discuss how these RFs properties relate to those reported in primates.

DISCUSSION

11. It is not trivial to relate properties of DNNs to those of biological systems. A key aspect of the primate visual system ignored here regards the key role of the fovea in high-acuity and color vision. This limitation requires further consideration. The authors may want to consider relate work emphasizing the importance of biological constraints on network properties such as learning rule, architectural constraints like cortical magnification of the fovea and hemifield representations.
12. Line 254: “[...] the greater heterogeneity in the parvocellular relative to the magnocellular pathway is in agreement with neurophysiological reports.” Network units and brain cells are different things. The question is in my view if this variability is

driven by the same factors. Again, as before when posed earlier in the manuscript, this argument still seems unclear to me. Making it easier to follow would facilitate evaluating the weight of the argument.

13. Line 267: Argument seems vague. Please flesh out and substantiate or consider removing.

14. Line 275: Regarding this point about invariance, please refer to comments on methods section, below.

15. Line 287: "Human studies revealing late-sighted children's deficits in global motion processing (Elleberg, 2002) also attest to the susceptibility of the magnocellular pathway to early visual deprivation, considering the task's alignment with magnocellular characteristics." This may be true, but it does not seem to speak to shape processing of static images as probed by the authors, but a completely different task. The authors should in my view provide a more balanced evaluation of literature both in favor and in disagreement of their interpretations.

16. Paragraph in lines 304 to 313 seems to me potentially important to justify the authors approach. It may be worth appealing to something like this also in the introduction. Having said that, it is not clear to me how the cited work established that "[...] we found that initially low spatial acuity and color sensitivity at birth as well as initially low temporal frequency sensitivity during prenatal development helped instantiate extended receptive field structures (spatial in the visual domain; temporal in the auditory domain) and more robust performance profiles later in life." It would be important to further clarify this evidence and might suggest using it in the introduction to motivate this study if the evidence is strong.

METHODS

1. Reference to "clusters" in Figures 1-4 (and, e.g. Line 250: "clear cluster") in my view benefit of more objective criteria for validation.

2. Line 329-331: I am not sure I follow the logic of this analysis decision. What would a noisy RF look like, and how would reducing the number of RFs help on this regard?

3. I feel that it is hard to judge the validity of the author's metric choices given the description provided in the method's section, which also does not include links to the actual code used to implement them.

In particular, I find it uninformative to simply indicate in lines 391-392 that "Finally, note that the metric is independent of the absolute strength of the signal, rendering the question of whether to use a normalized or unnormalized spectral decomposition obsolete" with no associate citations, access to code, or further derivations to demonstrate this point. While the authors may well be correct, I am skeptic of this claim and feel it would be important to clarify this issue to evaluate the validity of the chosen metrics.

Minor concerns

Introduction:

0) Lines 58-59: possible typo. Why jointly encode the two attributes if acuity and color sensitivity are weak/absent?

1) Line 67: The authors may want to consider additional relevant reviews (e.g., Serre, 2019) and work showing the influence of architectural constraints on the interpretation of network response patterns (e.g., Revsine et al., 2024).

2) Terminology line 69: manipulation of sensory experience or do the authors mean sensory input?

3) Lines 107-109: Statistical test may be unwarranted (strongly skewed distributions)

4) Line 125: "[...] we observe a clear cluster of magnocellular-like receptive fields [...]" The claim that there is a cluster has not been substantiated by objective criteria. It would be also important to clarify in what sense these RFs are not M-like.

5) Figure 2C: what is different about cars & boats?

6) It may make sense to change x-axes in Figure 3 to: Correct decisions (%), and indicate chance-level in figures, as well as construct & report confidence intervals.

7) It would be helpful to read more (perhaps in the SI) about how the authors impute decisions between shape-based/not texture based, etc.

8) Terminology: to my understanding RF's are not ablated. Units (or neurons) with RFs are ablated. E.g., see Figure 3 axes & text.

9) Terminology: Caption Fig 3 line 204: colorful first-layer receptive fields. Perhaps color-sensitive works better?

10) Line 216: what does this mean? "Indicative of the magno stream's greater temporal dynamics"

11) The authors focus on finding similarities between the literature and their observations. It would be probably good to focus

a bit more on differences.

12) Line 295-302: In my view this speculation needs to be further clarified. It may be useful to distinguish optical properties from neural properties given the aims of this study.

13) I am not sure what is the benefit of providing teleological perspectives in this context. Please clarify.

Reviewer #2

(Remarks to the Author)

Strong Points:

- * Innovative approach using multiple training regimes, including mixtures of blurred and color inputs.
- * Highly relevant and insightful analysis of receptive field properties in DCNN units, with a notable inclusion of motion studies.

Minor Points:

- * reference work on k-cells (e.g., VA Casagrande, TINS 1994), which might also enrich your discussion.
- * Clarify the rationale for using 48 receptive fields of 22x22 in layer 1 instead of the default network configuration, especially if results are similar. Why deviate from the standard?
- * What is the next convolutional layer doing? Could that be relevant?
- * Explain the choice of 200 epochs with a constant learning rate. Why? I do not think this is the SOTA training for Alexnet.
- * Include visualizations of loss functions for all networks to aid comparison.

Major points:

- * Robustness of results: Only one trained version of each network is presented. To establish reliability and rule out random drift, it is crucial to train multiple versions (e.g., 5) with different initializations. Present means with standard errors.
- * Discuss the methods to characterise the units (color, orientation, RF size) in relation to existing literature on this topic. How does this approach compare.
- * Discuss how this work relates to other papers looking at smoothing of stimuli used for training (for instance Frank Tong's work).

Version 1:

Reviewer comments:

Reviewer #2

(Remarks to the Author)

I want to thank the authors for their very extensive, and adequate replies to my queries. With the extensive work done for me (reviewer 2) and also looking at the work that has been done for reviewer 1 I think the paper improved in quality substantially. The first version of the paper was a joy to read and this has only improved.

I am left with only 1, more minor, question. that should be commented upon somewhere in the paper: as expected from the loss in Fig S21 the biomimetic networks take longer to reach a solid performance level. While not unexpected (and explaining the 200 training epochs) this is important to comment on.

Responses to reviewer comments

(Reviewer comments in blue; author responses in black)

Summary of changes in response to the reviewers' recommendations:

We would like to thank the reviewers for their insightful and helpful comments that helped us significantly improve our paper. In addition to our detailed point-by-point responses below, we wish to highlight several key changes to the paper. They include the following:

Introduction:

- We have extended the description of known spatial and temporal characteristics of parvo/magno cells.
- We have added a paragraph about the linkage between the magnocellular system and global processing.
- We have expanded the paragraph about past computational work on 'adaptive initial degradations' and moved it from the Discussion to the Introduction section.
- We have added a paragraph to preface our overall rationale. We also describe why greater heterogeneity among parvo cells is expected and generally explain our logic more clearly.
- We have provided more context for the use of deep networks as computational models.
- We have clarified the motivation for the (now 5) different training settings used.

Results and Supplementary Figures:

- For all trained networks, we repeated the training with 5 random initializations.
- We introduced three new network settings to test the generalizability of our findings. These are: 1. Using the original AlexNet architecture with 96 receptive fields, each of size 11x11 pixels; 2. Training with only half the epochs; 3. Using a decaying learning rate.
- We have included ablation results with low vs. high spatial frequency filters (in addition to ablation of low vs. high color filters).
- For the 3D-CNNs, we have also visualized receptive fields with minimal temporal variation.
- We have visualized training/validation loss and accuracy for all models trained.
- We have clarified and contextualized the results and have made the text more conservative. We have also linked it more to the Introduction section.

Discussion:

- The discussion of results is now more detailed, elaborated, and nuanced.
- The discussion of past computational work, such as Jang & Tong (2024), has been extended.
- Two paragraphs related to corroborating results from the experimental literature (as well as one paragraph describing discrepancies) have been added.
- Four new paragraphs detailing limitations / considerations have been added. They articulate: (i) the use of deep networks as computational models, (ii) the role of architectural constraints, (iii) differences between the receptive field properties observed in our study and those present in the real biological system, and (iv) the nature-nurture debate.

Methods

- We have provided more details concerning the different training settings and regimens, the texture/shape bias analysis, and invariance methodology, along with our overall rationale.
- Our metrics are described in greater detail and linked more to past literature. In addition, an analysis has been added regarding the reliability of the color metric.
- All code and data are now shared in a GitHub repository. This repository will be made permanent following acceptance.

Reviewers' comments:

Reviewer #1 (Remarks to the Author):

This decade, DNNs have been increasingly explored as models of primate vision. Many such explorations use popular network architectures such as AlexNet (e.g., used in this study) and train these models on categorization tasks with massive image databases like ImageNet (as done in this study). Imposing biologically relevant constraints on these models is relatively rare at the level of their architecture, learning rules, as well as the “visual diet” or “regimen” used to train them. The specific simulation study reviewed here explores the consequences of manipulating the “training regimen” (or “visual diet”) on receptive field (RF) properties of network units. The study emphasizes that a “biomimetic” training regimen (first, consisting of low spatial frequency & achromatic images, next of standard chromatic images also including higher spatial frequency information) leads to network units revealing response properties that mimic those of neurons recorded from the mammalian visual cortex—i.e., magno- and parvocellular responses. The authors claim the observation of such rough correspondence among units of DNNs and those of cortical visual neurons supports their hypothesis that “[...] the temporal confluence in the progression of spatial frequency and chromatic sensitivities during development may significantly shape neuronal response properties characteristic of this division.” The authors further make two ancillary claims, that (i) “[...] biomimetic training induces a more human-like classification bias towards global shape” and (ii) their findings have “relevance for the design of training procedures for computational vision systems”.

In my view, the paper is well written and interesting in the sense that, like few previous studies (more on this below), it could help redirect attention in the field to the importance of biological constraints and network training regimens when it comes to interpret DNN performance measures and response properties of network units. I think the authors make a good case showing the importance of the statistical structure (both spatial and temporal) of images to achieve more human-like biases and physiological-like response properties of network units. In other words, their observations in my view support the second clause of the final sentence of their abstract in lines 23-25: “[...] and applied relevance for the design of training procedures for computational vision systems.” The reported analyses also support the claim that biomimetic training led to a human-like global-shape bias. However, I am not convinced the currently presented simulations and analyses convincingly support the first claim in that same sentence that: “These results have implications for understanding a key aspect of visual pathway organization”, or that the authors fully articulate an account of the emergence of the parvo-magno- distinction. I am therefore also not convinced the presented RF analyses provide “strong support” for their main hypothesis.

We very much thank the reviewer for highlighting the potential significance and novelty of our study and for finding our work to be of interest. We greatly appreciate the many constructive suggestions, which have improved our paper substantially. Detailed responses to the helpful list of major and minor comments are provided further below, but we highlight some of the key points here.

Specifically, we acknowledge that in aiming for conciseness, the previous version of our manuscript may not have fully articulated our account of the emergence of the parvo-magno distinction. As described further below, we have now significantly expanded the exposition of our account in the Introduction section. This includes additional detail, context, and discussion related to spatial and temporal receptive field properties of magno- and parvocellular systems,

the link between the magnocellular system and global shape processing, past computational studies on “adaptive initial degradations”, early visual development, and a clearer explanation of our rationale and computational approach. Our expanded introduction better motivates the study and makes subsequent Result and Discussion sections (which have been significantly revised as well) easier to follow.

We also agree that our results do not provide definitive proof but rather add plausibility to the proposed developmental account of the parvo-magno organization. To reflect this more appropriately, we have revised our language and statements throughout the manuscript to be more nuanced. Further, as described in response to specific major/minor concerns further below, we have also added four new paragraphs to the Discussion section that address relevant considerations and limitations of the study. These are concerned with (i) the use of deep networks as computational models of the biological system, (ii) the role of architectural constraints, (iii) differences between the receptive field properties observed in our study and those present in the real biological system, and (iv) the nature-nurture debate.

Although multiple aspects of the reported research are in my view creative and insightful (e.g., manipulating interpretable low-level aspects of the visual diet, as well as some ensuing RF analyses), and the figures well-crafted and informative, I am concerned the logic of the paper seems somehow circular, and the information in the methods section in my view insufficient to confidently evaluate the validity of the chosen metrics; especially given the lack of access to the code that implements them. The authors aim to explain the emergence of units in DNNs with magno- vs parvo-cellular properties by manipulating properties of the images (chromaticity and blurring) that do not seem to be independent from the RF-properties of those neurons whose emergence the authors are trying to explain.

We are glad that the reviewer finds the research to be creative and insightful. As alluded to above, and as further detailed below, we now provide a clearer account of our rationale in the Introduction section, which, we believe, helps better motivate our approach taken.

We appreciate the specific concern of circularity. There are two observations that are worth highlighting here. First, while it might not be overly ‘surprising’ that biomimetic training leads to a bias in first-layer filters toward lower spatial frequency and color tuning, it is nevertheless an important demonstration. It would have been conceivable, for instance, that the filters of the biomimetic model learned in the first half of training are entirely overwritten in the second half. We express the latter in our revised Results section as follows:

“This effect highlights the significance of spatially extended and luminance-based receptive fields instantiated during the first half of training. Notably, these differences in RF sensitivity distributions persist notwithstanding the second half of the training with high spatial resolution and high chromatic content stimuli.”

Second, we are not just examining individual distributions of spatial frequency and color sensitivity but also joint ones. What emerges in the biomimetic model, but not the standard one, is a population of magnocellular-like receptive fields exhibiting *joint* tuning for low color and low spatial frequency. This joint tuning is not a necessary consequence of the biomimetic model having a lower color tuning distribution and, separately, a lower spatial frequency tuning distribution. Our new text in the Results section underscores that if color and spatial frequency

features were learned independently, we may not have observed joint tuning in the biomimetic model:

“The strong joint coding observed in these units is especially noteworthy given that it is not a necessary consequence of both color and spatial frequency distributions being tuned to lower values; they nevertheless could have emerged to be independent.”

Regarding the metrics and other computational analyses, we have now made the code available, added greater detail to the Methods section, and, as detailed in response to reviewer 2’s comments, further elaborated on, and tested, the metrics chosen.

The simulations explored also seem to ignore the fact that parvo-cells are highly concentrated at the fovea, and the primate ventral visual processing stream (believed to be engaged in shape representation and object recognition) exhibits a marked foveal response bias. This is an issue the authors should in my view consider, as well as relevant related work.

Thank you for raising this important point. As detailed in response to major concern 11, we have added a full paragraph to the Discussion section explicitly acknowledging the role of architectural constraints, including foveal/parafoveal representations.

As for the specific point about dorsal vs. ventral processing, it is important to note that recent work, as comprehensively reviewed in Ayzenberg & Behrmann (2022a, 2023), has highlighted the involvement of the dorsal pathway and its magnocellular input in global shape processing. We have significantly elaborated on the linkage between magnocellular units and global shape processing in the Introduction section. This is further elaborated on in response to major concern 3.

In sum, although the paper is in my view interesting and contributes creative analyses and a fresh perspective that might positively influence the field, I feel that the logic underlying key arguments supporting their central hypothesis may require the examination of some main claims in the current version of the manuscript. Specifically, I am not convinced the authors provide an account of the emergence of the parvo- and magnocellular systems based on early sensory development, since it would require disentangling the influence of magno and parvo RF properties on the image transformations used to implement the “biomimetic” training regime-- an admittedly very challenging task. The issues here are in my view reminiscent of the nature vs nurture debate. Both nature and nurture probably play a role, but unclear to me from reported simulations how to advance towards answering such questions.

We appreciate this important point. Indeed, it is possible that an interplay of “nature” and “nurture” may underlie parvo/magnocellular pathway formation. We have added a paragraph to the Discussion section highlighting that while our results demonstrate the possibility of the emergence of a parvo-magno distinction in a fairly generic deep network primarily based on developmental experience (“nurture”), it does not rule out the potential role of “nature”, or a combination of both, in biological systems.

It is important to note that the study also builds on the track record of past computational studies that have shed light on other nature/nurture debates in development. For instance, in their influential 1996 Nature paper, Olshausen & Field demonstrated that a type of learning algorithm finding sparse codes for natural scenes develops receptive fields exhibiting some

similarity to those reported in V1. While not ruling out the possibility of some innate biases, and despite some studies pointing to the presence of receptive fields in visually naive animals, the study demonstrates one way in which such receptive fields could emerge through experience, with important implications. We have summarized this in a second paragraph.

We have added the following two paragraphs to the Discussion section:

“A fourth consideration relates to the complex interplay of ‘nature’ and ‘nurture’. Our results show that, in principle, a generic visual system, when exposed to a biomimetic training progression, can develop a rough division of magnocellular-like and parvocellular-like units without the need for hard-coded separate pathways ab initio. In other words, our results demonstrate the possibility of an experience-driven (‘nurture’) route to such differentiation. At the same time, this demonstration does not rule out the possibility that, in the real biological system, some aspects of parvocellular and magnocellular organization might already exist through phylogenetic dispensation (‘nature’) prior to patterned sensory inputs (see also Rakic, 1977 and Gottlieb et al., 1985) and may be refined by subsequent experience. Attesting to this complex interaction, Crair et al. (1998) have reported that although some basic structures of cortical maps can form without actual sensory experience, the emergence of specific features of such maps is critically experience-dependent. It is also possible that both ontogenetic development (i.e., development within an individual’s lifetime, driven directly by sensory experience) and phylogenetic development (i.e., the evolutionary shaping of neural architecture and biases across generations, possibly driven over the long term by statistics of the sensory environment) play complementary roles in establishing and refining the parvo- and magnocellular pathways.

Despite some of these open questions, our findings build on previous computational studies demonstrating that certain aspects of the biological visual system can be reproduced without reliance on any explicit innate biases. For instance, Olshausen & Field (1996) showed that receptive fields exhibiting some resemblance to those of the mammalian primary visual cortex can emerge through sparse coding on natural images. More recently, Orhan & Lake (2024) reported that deep learning models without domain-specific, innate biases can develop sophisticated internal models of the world when exposed to child-like inputs. Similarly, Pandey et al. (2024) demonstrated that newborn chicks and generic deep networks, when exposed to the same impoverished environments, can learn similar object recognition abilities – supporting an experience-driven account of development. Collectively, these studies illustrate the power of computational modeling in informing longstanding nature-nurture debates (Wood, 2024). While not ruling out innate constraints, they underscore how experiential factors are capable of giving rise to representational organization, with potential developmental, computational, and evolutionary implications.”

Below, I provide a more detailed list of major and minor concerns. If the authors addressed these concerns, this might in my view be an interesting and potentially impactful contribution.

Thank you again for these helpful comments, which have improved our paper substantially. We respond in detail below.

Major concerns

1. Properly reviewing this paper would benefit from access to the code, images, and network models. In particular, it seems to me hard to evaluate the validity of some of the chosen metrics given the provided information (see comments to methods section, below).

Thank you for this important suggestion. As now detailed in our Methods section, we provide access to our code and data in the following repository: <https://github.com/marin-oz/parvo-magno>, which will also be permanently archived on Zenodo upon acceptance. In addition, we have expanded the relevant Methods sub-sections to better introduce our metrics (as detailed further below).

2. Line 129: It is unclear for me why larger heterogeneity is observed for units tuned to high spatial frequency OR chromatic content. Also unclear if the OR is inclusive or exclusive and, whichever it is, why.

We now describe in the Introduction section that while magno cells tend to jointly exhibit low color and low spatial frequency tuning, there are two subpopulations of parvo cells: those with higher spatial frequency tuning (and strong orientation selectivity) and others with lower spatial frequency tuning (and weaker orientation selectivity). This yields naturally greater heterogeneity among parvocellular-like units. The text added to the Introduction reads:

“It is worth prefacing here that the well-separated parvo and magno layers in the LGN partially mix in the visual cortex (Hubel & Livingstone, 1990). Consequently, two subpopulations of parvocellular units have been reported in the literature, both of which are color-sensitive to some extent: those exhibiting higher spatial frequency selectivity and typically strong orientation tuning (‘interblobs’), as well as those exhibiting lower spatial frequency selectivity and mostly low orientation tuning (‘blobs’) (Hubel & Livingstone, 1990). Given these sub-specializations that are plausible in the context of richer visual inputs, we would expect a greater heterogeneity of receptive field tuning in the parvocellular system, relative to the magnocellular one.”

In addition to minor rephrasing in the Results section as well as more conservative language to discuss the match between the biomimetic receptive fields and the two specific subpopulations of parvo cells, we also revised the Discussion section as follows:

“Biomimetic training also led to parvocellular-like receptive fields. However, rather than all being tuned to high spatial frequencies and color content, these units exhibited greater heterogeneity in terms of their tuning profiles. As described in the introduction, in light of the two subpopulations of parvocellular units (blobs and interblobs) observed at the level of the primary visual cortex (Hubel & Livingstone, 1990), it is interesting to note the analogous heterogeneity among parvocellular-like units in the simulations.”

3. Line 143: It is unclear for me why the narrative of the paper shifts from M-P story to global shape & the Gheiros story. It is also unclear why the authors link global shape processing to magno-cells, and not parvo-cells. Further motivating these issues in the introduction as well as relevant references might help clarify this point.

Thank you for highlighting the need to elaborate on the global-shape narrative early in the manuscript. We have added a dedicated paragraph to the Introduction section in order to motivate this link, citing experimental evidence indicating an important role for the magnocellular system and dorsal pathway in global shape and configural processing. This helps motivate the subsequent analyses reported in the Results section. The added paragraph reads:

“Drawing on these spatial and temporal characteristics from neurophysiology, along with early psychophysical and clinical studies (e.g., Livingstone & Hubel, 1987; Livingstone et al., 1991), the magnocellular system has been assigned “a more global function of interpreting spatial organization” (Livingstone & Hubel, 1988). Its role has also been likened to the processing of Gestalt cues for object understanding and linking properties (Livingstone & Hubel, 1987; 1988). In contrast, the parvocellular system has been implicated in the analysis of greater spatial detail (Livingstone & Hubel, 1988). Recent human neuroimaging studies have more directly attested to the involvement of areas in the dorsal visual pathway, which are driven by strong magnocellular input, to global processing (see Ayzenberg & Behrmann, 2022a, 2023 for comprehensive review). For instance, dorsal areas of the visual system were shown to be crucial for the analysis of the broader spatial arrangement and global shape of object parts (Ayzenberg & Behrmann, 2022b) as well as for the configural processing of faces (Zachariou et al., 2017). Disrupting processing in these areas using transcranial magnetic stimulation was furthermore shown to lead to a specific impairment of global, not local, processing (Romei et al., 2011). Thus, even though both parvo- and magnocellular systems are likely to contribute to complex behavior, and their individual contributions cannot be entirely disentangled, these studies speak to the significance of the magnocellular system in global processing.”

4. Line 159: “we chose low color, rather than low spatial frequency as a proxy for M-units considering the larger homogeneity of the former, [...]”. This seems to me arbitrary. How do these analyses look if spatial frequency is chosen as proxy for M-units. Similar comment extends to line 165 regarding the causal role of RFs exhibiting magno characteristics.

Thank you for this suggestion. We have now included analogous analyses using spatial frequency, revealing qualitatively fairly similar results, albeit marginally weakened in terms of the absolute effect strength. For the main network setting used, these new analyses are shown in Supplemental Figure 18. They are also provided for any of the other training settings in Supplemental Figures 13-17. The revised Results section reads:

“To this end, we gradually eliminated (or ‘ablated’) the most color-tuned (i.e., more parvo-like) vs. the least color-tuned (i.e., more magno-like) first-layer receptive fields of the trained networks and re-computed the shape bias. ” [...]

“Repeating the same experiment by ablating the receptive fields most tuned to high frequencies (i.e., more parvo-like) vs. low frequencies (i.e., more magno-like) reveals qualitatively similar results, albeit marginally weaker (Figure S18).”

The color metric had initially been the primary choice as individual RFs appeared more spread in terms of color metric values, compared to spatial frequency metric values, and were therefore more separable. Nonetheless, carrying out the analogous analysis with spatial frequency yields comparable outcomes.

Shown below is Figure S18 (depicting the results for setting 1):

Fig. S18. Results of unit ablation on shape/texture bias of setting 1 when carried out both with regard to the color and frequency metrics. The shape/texture bias is depicted as a function of the proportion of ablated units with lowest color, highest color, lowest frequency, and highest frequency (from left to right) tuning. Depicted are the five training runs (thin lines) as well as their means (thick lines).

Also included here is Figure S13C&D (depicting the results for setting 2):

Fig. S13. Reproduction and extension of Figure 2 in the main manuscript when utilizing setting 2 (96 22x22 pixel RFs). **C&D.** Shape/texture bias as a function of the proportion of ablated units with lowest color, highest color, lowest frequency, and highest frequency (from left to right) tuning. Depicted are the five training runs (thin lines) as well as their means (thick lines).

In both settings, ablation of low color / frequency filters reduces the shape bias of the biomimetic model much more substantially than ablation of corresponding high color / frequency filters.

5. Line 177: Unit ablation and invariance studies. Method to assess invariance incompletely clear, as well as how it may be influenced by possible normalizations (please, see comments to methods section further below)

In addition to providing access to our analysis code, we have expanded the Methods section with a detailed explanation of our invariance analysis, as follows:

“In Figures 3C&D, we present an invariance analysis aimed at evaluating how stable a network’s unit activations are when input images are manipulated. Specifically, we computed distributions of correlation coefficients by comparing network activations produced by full-color, non-blurred images against activations produced by the same images when presented either in grayscale (Figure 3C) or when blurred (Figure 3D). Our goal was to examine whether the biomimetic model, compared to the standard network, maintains more similar responses under these alterations.

For computational efficiency, we began by subsampling the ImageNet test set. While the original test set comprises 50,000 images (50 images per ImageNet class), we randomly selected three images per class, producing a dataset of 3,000 test images, which was used for all subsequent invariance analyses. Each of these 3,000 images was then fed into both the biomimetic and standard networks in three forms: (1) as undegraded images (i.e., in full-color and non-blurred), (2) converted to grayscale, and (3) blurred with a Gaussian filter ($\sigma = 4$).

*For each of the networks, each of the 3,000 test images, and each of its three variants (undegraded, blurred, and grayscale), we extracted the activations of every unit across all convolutional and fully-connected layers. We then concatenated these activations across the 3,000 images for each of the three variants. As an example, consider the first convolutional layer of our minimally-adjusted AlexNet in setting 1. This layer has 48 filters, and for each filter, the activations corresponding to each of the three image variants can be represented by a $22 * 22 * 3 * 3000$ matrix (height * width * number of color channels * number of test images). Each matrix was then flattened into a single 1D vector, such that every filter produced three vectors – one each for undegraded, grayscale, and blurred images.*

Using these vectors, we computed correlations to quantify the similarity of responses across image alterations. For each unit (e.g., each filter in the first convolutional layer), we correlated its undegraded-image activation vectors with its grayscale-image activation vector and, separately, with its blurred-image activation vector. Figures 3C and 3D show the distributions of these correlation values for all network layers. In the example of the first convolutional layer, each distribution comprises 48 correlation values – plotted in blue for the biomimetic model and in red for the standard model. Similar distributions were computed for all other convolutional and fully-connected layers.”

Regarding the specific point related to normalization, it is worth highlighting that the key computation performed is a correlation, which would not be affected by simple normalization.

Moreover, we are not carrying out a normalization prior to feeding the activations into the correlation computation. We believe that the more detailed methods description helps explain this procedure more clearly.

Moreover, we have slightly revised the Results section, as described in our response to the next major concern (# 6) below.

6. Line 180-189 “[...] Further, the relative importance of these units remains more similar across full-color vs. grayscale images for the biomimetic network than for the standard network, presumably due to increased invariance to the removal of chromatic information. To quantify the invariance of both networks to the removal of chromatic and high spatial frequency content, [...]. *This analysis reveals markedly greater invariance of the biomimetic model for both stimulus dimensions across nearly all network layers.*” (italics mine for emphasis)

The basic rationale of this analysis is that we expect the network relying more on color or high spatial frequency to show greater changes in activations under grayscale or blurred conditions. Thus, if a network (like the biomimetic one) uses more magnocellular-like features, we would anticipate more stable activations even when removing color or high spatial frequencies. In addition to a more detailed description in the Methods section (see first paragraph, as included in response to major concern 5 above), we have revised our Results section to make it easier to understand, as follows:

“Further, as is evident through comparison of the top vs. bottom panels of Figures 3A&B, performance curves remain more similar across full-color vs. grayscale images for the biomimetic network than for the standard one. This effect may be due to increased invariance of the biomimetic model to the removal of chromatic information. To quantify the invariance (i.e., maintained similarity of activations) of both networks to the removal of chromatic as well as high spatial frequency content, we computed the distribution of correlation coefficients across all units for activations elicited by color vs. grayscale images (Figure 3C) and full-frequency vs. blurred images (Figure 3D) (see Methods for details). For both stimulus dimensions, the obtained distributions lean towards higher correlation values for the biomimetic than for the standard model, suggesting greater invariance of the biomimetic model.”

7. Line 191:Analyses of invariance to the reduction of image contrast. It is not clear to me why the authors decided to explore contrast responses, or the validity of their analysis approach (see comments on methods section, below)

Thank you for raising this point. We respond together with point 8 below.

8. Lines 196-200: “This observation is aligned with the neurophysiological finding that magno cells have greater contrast sensitivity and thereby saturate with lower contrast (Shapley et al., 1981; Kaplan & Shapley, 1982; Hicks et al., 1983; Derrington & Lennie, 1984), *which would be expected to result in higher correlations between neural units across normal and low-contrast stimuli.*” (italics are mine for emphasis) This is to my understanding not necessarily true. Correlations need not change as a function of contrast even if responses revealed substantial contrast sensitivity. It seems also vague what the authors mean by “normal” and “low-contrast”

stimuli. It would be helpful if the authors clarified the assumptions of their argument. A simple supplementary simulation may (or may not) help to clarify their point.

In light of the numerous very helpful suggestions from both reviewers, whose incorporation led to considerable lengthening of our manuscript (and quadrupling of our supplementary figures), we have opted to remove Figure 3E and its discussion of contrast. While potentially interesting, it is much less central to our main arguments. We therefore decided to maintain a tighter focus on the key examinations carried out in this paper, and provide more context, details, interpretation, and discussion related to these key examinations. We believe that this change has improved the flow of our paper.

9. Line 213: Summary of analyses of temporal characteristics of magnocellular units. “The investigations above revealed that biomimetic training leads to the emergence of a group of receptive fields that exhibit consistent magnocellular-like characteristics in terms of chromatic and spatial frequency tuning as well as their functional roles in classification behavior.” Why do the authors believe they have probed and can relate the functional role of their magno-like units and real magno-cells? Is there evidence that real magno cells exhibit the functional properties described here? References and a more thorough discussion of this matter seems important.

As described in response to major concern 3, we have elaborated more on the functional role by expanding our Introduction section and emphasizing the magnocellular system’s involvement in global shape processing.

In addition, as further detailed in response to major concern 12, we now explicitly describe in the Discussion section differences between the magno-like units observed in our simulations and real magnocellular neurons in the biological system.

Thank you again for having brought up these important points.

10. Regarding the reported spatiotemporal RF analyses, it would be useful to compare results with similar analyses for the low temporal variation & low color units in the biomimetic and standard models. It would be also informative to report the distribution of RF sizes of network units, and discuss how these RFs properties relate to those reported in primates.

Thank you for highlighting the need to extend the spatiotemporal receptive field figures as well as the interpretation thereof. We have done so on several fronts. First, to aid comparison, we now also include the receptive field scatter plot (depicting the relationship between temporal variation metric, spatial frequency tuning, and color tuning) of the standard model directly in the main manuscript (Figure 4F). Second, we visualize the most temporally varied receptive fields of the standard network as part of Figure 4G as well. Third, we now also visualize the *least* temporally varied receptive fields of both models as part of Supplementary Figure 20. Finally, in response to reviewer 2’s helpful suggestion to repeat all training runs with (here, 5) different random initializations, Supplementary Figure 19 depicts the results of all 5 training runs.

The updated Figure 4 is included below:

Figure 4. A&B. Histograms of color (A) and spatial frequency (B) metrics for receptive fields of the biomimetic and standard model when training is conducted with video inputs to a 3D-CNN. **C.** Relationship between temporal RF properties (using the temporal variation metric; see Methods) and spatial RF characteristics (using the color and spatial frequency metrics used before) of the biomimetic model. The ellipse marks the five receptive fields exhibiting the greatest temporal variation. These RFs also show consistently low color and spatial frequency tuning. **D.** Visualization of the five most temporally varied receptive fields of the biomimetic model. **E.** Depiction of biphasic receptive field, adapted from the neurophysiological study by De Valois et al. (2000). **F.** Relationship between temporal variation metric and spatial RF characteristics (color and spatial frequency metrics) of the standard model. The ellipses mark the five receptive fields exhibiting the greatest temporal variation. **G.** Visualization of the five most temporally varied receptive fields of the standard model. All plots in Figure 4 have been generated on the basis of the first out of five training runs with different random initializations. Results of all five training runs are shown in Figure S19.

Figure S19 is also included below, depicting the least temporally varied receptive fields for each of the two regimens:

Fig S19. Visualization of the five least temporally varied receptive fields of the standard and biomimetic models (based on the first out of five training runs).

The suggested consideration of low temporal variation RFs pointed to an important pattern in the data: Indeed, as pointed out before, the most temporally-varied RFs (in the biomimetic model; to some extent also in the standard model) exhibit spatially-magnocellular response properties. However, considering that there are few RFs with strong temporal variation, there do exist several spatially-magnocellular units that do not exhibit strong temporal variation. The revised Results section reads as follows:

“This analysis reveals that units exhibiting maximal temporal variation in the biomimetic model (i.e., units aligned with magnocellular properties in the temporal domain) indeed show consistently strong magnocellular characteristics in the spatial domain (i.e., low chromatic and spatial frequency tuning) (Figure 4C; Figure S19 for other training runs). Such units, depicted in Figure 4D, exhibit strong similarity to biphasic cells (illustrated in Figure 4E), which, as introduced earlier, are predominantly found in the magno pathway (De Valois et al., 2000). It is important to note, however, that most receptive fields generally exhibited relatively low temporal variation (possibly due to the task requiring only moderate incorporation of temporal cues for classification success). Thus, while the most temporally varied receptive fields in the biomimetic model consistently exhibit spatially-magnocellular response properties, there are several spatially-magnocellular units that do not exhibit strong temporal variation. The receptive fields exhibiting the least temporal variation are depicted in Figure S20. Interestingly, for a 3D-CNN trained with the standard regimen, several of the most temporally varied receptive fields tend to also be associated with relatively low-frequency, achromatic content (Figure 4F; Figure S19 for other training runs). However, this association is less clear-cut than for the biomimetic regimen, with some maximally temporal receptive fields also tuned to higher frequency or color content (Figure 4G).”

Regarding the specific point related to receptive field sizes, we would like to highlight that while the size of the filters is spatially fixed at 22x22 pixels, we do provide a characterization of spatial frequency sensitivity as well as a comparison between the two models (e.g., in Figure 4B), which can serve as an estimation of effective relative receptive field sizes in terms of the extent of an image that information is integrated across. As for receptive field properties in primates, we have, as described above, further expanded on past literature in the Introduction and Discussion sections. The general pattern is that magnocellular receptive fields have weak color and spatial frequency tuning but exhibit strong sensitivity to high temporal frequencies. These response properties exhibit similarity to those of the receptive fields inside the circle of Figure 4C (and, to a lesser extent, of Figure 4F; primarily, the top-2 temporally-varied receptive fields).

DISCUSSION

11. It is not trivial to relate properties of DNNs to those of biological systems. A key aspect of the primate visual system ignored here regards the key role of the fovea in high-acuity and color vision. This limitation requires further consideration. The authors may want to consider relate work emphasizing the importance of biological constraints on network properties such as learning rule, architectural constraints like cortical magnification of the fovea and hemifield representations.

Thank you for raising this important point. We have introduced two new paragraphs in the Discussion, featuring (i) challenges associated with the use of deep neural networks as models of the biological system and (ii) the role of architectural constraints:

“Several additional considerations and potential limitations merit discussion. The first is inherent in any computational study that relies on the use of deep neural networks. Despite the successes of deep networks in modeling certain aspects of visual processing, they still diverge from human vision in important ways. Bowers et al. (2023) highlight significant discrepancies between deep network behavior and findings from human psychology. Indeed, many studies have shown that DNNs differ from humans in such aspects as global shape encoding (Baker et al., 2018, 2022), Gestalt grouping (Biscione & Bowers, 2023), crowding (Doerig et al., 2020; Lonqvist et al., 2020), as well as generalization and robustness (e.g., Geirhos et al., 2018b). As Wichmann & Geirhos (2023) discuss, deep networks may be “promising” but not yet fully “adequate” models of visual processing. However, pinpointing mismatches between the biological and artificial systems can lead to insights into the former and iterative refinements of the latter (Doerig et al., 2023; Lonqvist et al., 2021). In the context of the present study, introducing developmentally inspired input progressions offers potential insight into the emergence of aspects of the biological system and illustrates one way in which known limitations of deep networks in spatially global processing could be mitigated.

A second limitation concerns architectural details. Our model does not incorporate lateral and/or top-down recurrence, which has been proposed to be important for challenging tasks involving occlusion or noise (Spoerer et al., 2017; Tang et al., 2018), for capturing the representational dynamics of the human ventral stream (Kietzmann et al., 2019), and for accounting for complex analyses beyond the initial feedforward sweep (Kreiman & Serre, 2020). Moreover, parvocellular neurons are known to vastly outnumber magnocellular ones in the fovea, and the cortical representation of the fovea is disproportionately large (reviewed in Strasburger et al., 2011). As demonstrated by Revsine et al. (2024), various architectural constraints, such as cortical magnification, convergent feedforward projections, and interhemispheric connections, can critically affect the interpretation of neural data. In the current study, we chose a simplified approach that does not incorporate these constraints, in part because such constraints are known to change throughout development. For instance, feedback connections are reported to emerge a few months later than feedforward ones (Rockland & Pandya, 1979; Berezovskii et al., 2011), rendering the architecture of the initial visual experience more feedforward. Similarly, the fovea is notably more immature at birth than the parafovea (Abramov et al., 1982; Hendrickson & Drucker, 1992), which, too, may significantly affect neural encoding. Given these complex developmental factors, any architectural constraints incorporated may need to undergo changes throughout

training, which presents its own challenges. While these important considerations lie beyond the scope of the present paper, we acknowledge that they could influence developmental outcomes and warrant several detailed follow-up investigations.”

12. Line 254: “[...] the greater heterogeneity in the parvocellular relative to the magnocellular pathway is in agreement with neurophysiological reports.” Network units and brain cells are different things. The question is in my view if this variability is driven by the same factors. Again, as before when posed earlier in the manuscript, this argument still seems unclear to me. Making it easier to follow would facilitate evaluating the weight of the argument.

As described above, we attempted to make our narrative easier to follow throughout the manuscript. Moreover, we now explicitly acknowledge the differences between network units and brain cells. In addition to the two added limitations sections described in response to major concern 11 above, we have made two changes in response to this point. First, we revised the section (previously featuring “in agreement with neurophysiological reports”) with more neutral language, as follows:

“Biomimetic training also led to parvocellular-like receptive fields. However, rather than all being tuned to high spatial frequencies and color content, these units exhibited greater heterogeneity in terms of their tuning profiles. As described in the introduction, in light of the two subpopulations of parvocellular units (blobs and interblobs) observed at the level of the primary visual cortex (Hubel & Livingstone, 1990), it is interesting to note the analogous heterogeneity among parvocellular-like units in the simulations. ”

Second, we added a third limitations paragraph to the Discussion section, cautioning that observed distributions in CNNs do not replicate the full complexity of real visual pathways:

“A third limitation worth noting is that while our analyses focus on color sensitivity, spatial frequency, and temporal response characteristics, our learned receptive fields differ from those in real biological systems in several respects. For example, biological receptive fields often exhibit specific types of color opponency, whereas we adopt a simpler ‘more vs. less’ color metric. Moreover, while biomimetic training yields a population of magnocellular-like units and more heterogeneous parvocellular-like units, these distributions are not categorical and may represent a broader continuum. Finally, our analyses do not explicitly consider koniocellular (K) cells, which constitute a highly heterogeneous collection in the primate LGN and feature multiple subclasses of units with distinct response properties (reviewed in Casagrande, 1994 and Hendry & Reid, 2000). As Hendry & Reid (2000) note, the existence of this additional and highly heterogeneous pathway, alongside the parvo- and magnocellular streams, “introduces a complexity into a field only recently viewed as clean and simple”. Thus, while our approach captures some aspects of the organization in the mammalian visual system, a more comprehensive account will need to incorporate this additional complexity as well.”

13. Line 267: Argument seems vague. Please flesh out and substantiate or consider removing.

We have revised that passage to be more neutral and explanatory:

“It is worth noting that a 3D-CNN trained with the standard regimen yielded a similar pattern of results, although to a lesser extent. This points to an interesting difference between the spatial and temporal dimension. Specifically, in the spatial domain, biomimetic training was necessary to produce a differentiation between magnocellular-like (low color, low spatial frequency) and parvocellular-like (higher color, higher spatial frequency) units. In contrast, an association between low spatial and high temporal frequency tuning and, thus, a specialization to either fine-grained spatial features without much temporal variation or coarse-grained spatial features with stronger temporal dynamics appears present, to some extent, even when following the non-biomimetic training scheme.”

14. Line 275: Regarding this point about invariance, please refer to comments on methods section, below.

As described above, we have clarified the procedures for invariance measures in the Methods section.

15. Line 287: “Human studies revealing late-sighted children’s deficits in global motion processing (Elleberg, 2002) also attest to the susceptibility of the magnocellular pathway to early visual deprivation, considering the task’s alignment with magnocellular characteristics.” This may be true, but it does not seem to speak to shape processing of static images as probed by the authors, but a completely different task. The authors should in my view provide a more balanced evaluation of literature both in favor and in disagreement of their interpretations.

Thank you for this very important and helpful suggestion. We have significantly strengthened this argument and expanded our discussion of late-sighted children’s outcomes, citing multiple additional behavioral tasks. The new paragraphs (the last of which also discusses disagreements) read as follows:

“Our computational results obtained with the non-biomimetic regimen suggest that dispensing with initially degraded inputs characteristic of normal visual development reduces global shape processing as well as the presence of magnocellular-like units tuned to both low spatial frequencies and colors. Interestingly, this result is corroborated by past behavioral and neural data.

Behavioral evidence derives primarily from studies of individuals who were born blind and gained sight late in life through cataract surgery (for review, see Vogelsang et al., 2024b). Notably, these individuals differ from normally-sighted controls not only in their pre-surgical deprivation but also in terms of the quality of their visual experience post-surgery. As many of the maturational processes, which are significantly driven by retinal development (Banks & Crowell, 1993), continue despite the children’s blindness, their post-surgical visual system is markedly more mature than that of a newborn (Banks and Crowell, 1993, Wilson, 1993, Boas et al., 1969, Hendrickson and Boothe, 1976). Indeed, their behaviorally-assessed visual acuity, while below that of typically developed controls, is above neonate levels (Ganesh et al., 2014), and their color sensitivity reaches normal levels even just a few days post-surgery (Vogelsang et al., 2024a). In other words, the late-sighted commence visual experience with fairly high-quality inputs right from the start of their post-operative visual journey and thereby skip much of the initially degraded period that is characteristic of normal visual development. Our

simulations predict detrimental representational and functional consequences of such a start with abnormally high-quality input. Consistent with this prediction, late-sighted children were found to exhibit deficits in tasks requiring extended spatial processing, such as face recognition and configural face judgements (Geldart et al., 2002; Putzar et al., 2010; de Heering & Maurer, 2014; Vogelsang et al., 2018). Similarly, Ellemberg et al. (2002) reported deficits in global motion processing in the late-sighted, and McKyton et al. (2015) found that they exhibit intact local shape discrimination but impaired global shape completion and illusory contour perception. Vogelsang et al. (2025) also reported that relative to resolution-acuity-matched controls, late-sighted children exhibit reduced vernier acuity. As an instance of hyperacuity, vernier offset detection has been attributed to extended spatial organization and multi-unit cortical processing (Westheimer, 1979; McKee & Levi, 1987; Poggio et al., 1992). Finally, the late-sighted were found to show deficits in object recognition generalization to color-removed images (Vogelsang et al., 2024a). While we cannot rule out the possibility that some of these deficits could arise from deprivation during critical periods specific to certain visual functions (see, e.g., Geldart et al., 2002), the reduced access to early degraded inputs provides a parsimonious unifying account.

In addition to these behavioral linkages, neural evidence derives from reversible suture experiments in monkeys. Specifically, Le Vay et al. (1980) found that artificially induced early deprivation, followed by later restoration of sight, led to greater long-term damage to the magnocellular than to the parvocellular units in the visual cortex. In light also of the magnocellular system's stronger inputs and greater maturity at birth (Kennedy et al., 1985; Rakic et al., 1977; Gottlieb et al., 1985), its greater susceptibility to early deprivation has been accounted for by the earlier onset of its effective inputs. While these magnocellular deficits could, in principle, be accounted for by an anatomically hard-wired and pathway-specific critical period, following which, due to reduced plasticity, sight restoration does not allow the magnocellular pathway to gain normal function, an alternative account may be based on the lack of initially degraded inputs, preventing the magnocellular pathway from developing normally. Our computational simulations add plausibility to this proposal, considering that training with high-quality visual inputs from the beginning, as opposed to initial training on low-frequency, achromatic stimuli, resulted in the marked absence of receptive fields exhibiting magnocellular characteristics.

Although the above findings support the idea that the magnocellular pathway may be especially reliant on initially degraded visual input for its normal development, some experimental observations are not entirely consistent with this account. As reviewed earlier, the magnocellular pathway is associated with higher temporal frequency responses. Thus, one might expect late-sighted individuals to exhibit deficits in temporal sensitivity commensurate with their spatial deficits. However, empirical studies indicate that temporal vision is more resilient than spatial vision to early-onset visual deprivation (Ellemberg et al., 1999; Ye et al., 2021). Even in the spatial domain, although the late-sighted surpass neonates in terms of visual acuity, they still fall short of typically developed adults (Ganesh et al., 2014). These findings suggest a more nuanced developmental picture, in which certain aspects of magnocellular function may be less susceptible to atypical early experience and in which parvocellular functions are not entirely spared under such conditions.”

16. Paragraph in lines 304 to 313 seems to me potentially important to justify the authors approach. It may be worth appealing to something like this also in the introduction. Having said that, it is not clear to me how the cited work established that “[...] we found that initially low spatial acuity and color sensitivity at birth as well as initially low temporal frequency sensitivity during prenatal development helped instantiate extended receptive field structures (spatial in the visual domain; temporal in the auditory domain) and more robust performance profiles later in life.” It would be important to further clarify this evidence and might suggest using it in the introduction to motivate this study if the evidence is strong.

Thank you for this suggestion. We have moved this material to the Introduction section in order to motivate aspects of the “adaptive initial degradations” viewpoint before detailing the specific approach taken in the present study. It reads as follows:

“Recent computational investigations have begun exploring potential benefits conferred by such early degradations, although, thus far, they have focused on individual visual dimensions (e.g., acuity or color) in isolation.

In the domain of visual acuity, Vogelsang et al. (2018) have shown that training deep convolutional neural networks with inputs that transition from blurry to non-blurry leads to first-layer receptive fields tuned to lower spatial frequencies, effectively integrating information over more extended areas of the input image. This training progression, compared to several non-developmental ones, also yielded the best generalization performance across varying blur levels, suggesting potential benefits of commencing training with blurry inputs, at least in the domain of face recognition (Vogelsang et al., 2018; Jang & Tong, 2021). Although without an explicit developmental motivation, Jang & Tong (2024) recently also demonstrated that training a deep network with various levels of blurred inputs (including strong blur) can lead to a greater similarity with neural responses in macaque visual cortices, greater robustness to image perturbations, and an increased bias to classify images based on global shape rather than local features. Yoshihara et al. (2023) also report a slightly increased shape bias, but more modest than the strong-blur results reported by Jang & Tong (2024), when training with a mix of blurred and non-blurred images or when transitioning from blurred to non-blurred inputs. Separately, in the domain of color vision, Vogelsang et al. (2024a) found that a deep network training progression transitioning from achromatic to full-color inputs leads to greater robustness to later color removal or chromatic changes as well as to more stable, luminance-based receptive fields. Together, these studies highlight the potential benefits of sensory degradations early in life – a proposal we have referred to as the ‘adaptive initial degradation’ hypothesis (Vogelsang et al., 2024b). While this past work underscores how initial degradations might be advantageous for the developing system, it has, thus far, focused on isolated visual dimensions.”

METHODS

1. Reference to “clusters” in Figures 1-4 (and, e.g. Line 250: “clear cluster”) in my view benefit of more objective criteria for validation.

We have now removed the term “cluster” throughout the manuscript in favor of more conservative and descriptive language, such as “a magnocellular-like population of receptive fields exhibiting both low spatial frequency and low color tuning”. What we were referring to is the presence of many units of the biomimetic model that are jointly tuned to low spatial

frequency and color content. We do not attempt formal clustering but rather emphasize the apparent grouping in color-frequency-sensitivity space.

2. Line 329-331: I am not sure I follow the logic of this analysis decision. What would a noisy RF look like, and how would reducing the number of RFs help on this regard?

Before training, receptive field weights are randomly initiated and do not contain any systematic structures. Then, throughout training, receptive fields tend to specialize and form clear structures (for instance, Gabor-like patches tuned to a certain spatial frequency or blob-like structures responding maximally to a certain color, among others). These receptive fields can be meaningfully and reliably characterized through the use of metrics. If certain receptive fields were to not acquire any clear and meaningful structures but remain relatively “random” (or “noisy”), the reliability of analyses could potentially be reduced. Therefore, as a precaution, and to avoid over-parameterization, our first approach was to reduce the number of receptive fields to be learned (note that we had also increased the receptive field sizes from 11x11 to 22x22 in order to more precisely characterize spatial frequency sensitivity). With this smaller set of receptive fields to be learned, we expected that there would be an even stronger need for the network to build meaningful structures. Indeed, work with deep nets in a different domain – that of learning genomic sequence motifs – supports this expectation. Koo & Eddy (2019) have shown that the use of too many first-layer convolutional filters can lead to more filters not learning any meaningful representations.

It is important to note, however, that this was just a precaution and results obtained with the AlexNet equipped with the original set of 96 first-layer filters yields qualitatively similar results. In fact, in response to reviewer 2, we are providing, now in greater detail, results of simulations with 96 receptive fields, each of size 22x22, as well as with 96 originally-sized (11x11) receptive fields. The results across all of these settings are comparable.

To improve expositional clarity in the manuscript, we have now explained this modeling decision in the Introduction section, also highlighted that many other simulations were run (including those with the original set of 96 receptive fields), and used different phrasing to reflect that simulations across all of these settings are important. The revised Introduction paragraph reads as follows:

“To examine the generalizability of our results, these comparisons were carried out across several network settings, which are illustrated in Figure S1 in the supplementary material. In ‘setting 1’, for which results are reported in the main manuscript, we adapted the AlexNet to feature larger first-layer filters (increased from 11x11 to 22x22 pixels) in order to avoid restricting the types of receptive field structures to be learned and to allow for the extraction of frequency-based metrics with higher resolution. We also reduced the number of receptive fields from 96 to 48 as a precautionary measure to increase the likelihood of all receptive fields learning clear structures that can be meaningfully quantified through the use of metrics. In addition, we also report results with the full set of 96 receptive fields (size increased to 22x22 pixels) (‘setting 2’), with the full set of 96 normally-sized (i.e., 11x11 pixel) receptive fields (‘setting 3’), with only half of the training epochs (‘setting 4’), and with a decreasing learning rate (‘setting 5’). (...)”

In addition, we added to the Methods section:

“For setting 1 reported in the main manuscript, we slightly adapted the original architecture on two fronts. First, we enlarged the size of the receptive fields in the first convolutional layer from 11x11 to 22x22 pixels in order to avoid restricting the types of receptive field structures that can be learned by the network, and to allow for the extraction of frequency-based metrics with higher resolution and precision. Second, the number of first-layer receptive fields was reduced from a total of 96 to 48 as a precautionary measure to increase the likelihood of all receptive fields learning clear structures that can be meaningfully and reliably quantified through the use of metrics. (Although originating from a different domain, in the context of learning genomic sequence motifs, past work has shown that too many first-layer convolutional filters can lead to more filters not learning meaningful representations (Koo & Eddy, 2019).) (...).”

3. I feel that it is hard to judge the validity of the author’s metric choices given the description provided in the method’s section, which also does not include links to the actual code used to implement them.

As described above, all code is now publicly shared. Also, in response to some of the points brought up here as well as by reviewer 2, we significantly expanded the Methods section, including context and justification of our metrics (further detailed in response to a point raised by reviewer 2).

In particular, I find it uninformative to simply indicate in lines 391-392 that “Finally, note that the metric is independent of the absolute strength of the signal, rendering the question of whether to use a normalized or unnormalized spectral decomposition obsolete” with no associated citations, access to code, or further derivations to demonstrate this point. While the authors may well be correct, I am skeptical of this claim and feel it would be important to clarify this issue to evaluate the validity of the chosen metrics.

The metric is independent of any ‘absolute’ strength due to its denominator. For simplification, imagine the following two hypothetical vectors:

a(1) = 1; a(2) = 5; a(3) = 4.
b(1) = 10; b(2) = 50; b(3) = 40.

The metric, as described in the Methods section, is:

$$\text{weighted average frequency} = \frac{\sum_f \text{amp}(f) * f}{\sum_f \text{amp}(f)}$$

For a, what would be computed is: $(1*1 + 5*2 + 4*3) / (1 + 5 + 4) = 23/10 = 2.3$

For b, what would be computed is: $(10*1 + 50*2 + 40*3) / (10 + 50 + 40) = 230/100 = 2.3$

In addition to providing access to the code, we have clarified the role of the denominator in the Methods section:

“Note that our metric, due to its denominator dividing by the total sum of amplitudes, is independent of the absolute strength of the signal (...).”

Minor concerns

Introduction:

0) Lines 58-59: possible typo. Why jointly encode the two attributes if acuity and color sensitivity are weak/absent?

As alluded to above, we now explain our rationale more up-front, as part of the Introduction section, as follows:

“Prefacing our rationale detailed further below, we examine whether the tuning of some units (magnocellular-like) to both low spatial frequency and low color content could potentially be due to the degraded color and frequency characteristics of visual inputs early in life, with parvocellular-like units being more optimized for processing high-fidelity inputs that become available later in development. To this end, we test here whether a relatively generic deep convolutional neural network, when exposed to a developmentally-inspired trajectory of training inputs, is able to recapitulate key aspects of parvo- and magnocellular organization in terms of receptive field properties and functional correlates in terms of global shape processing.”

1) Line 67: The authors may want to consider additional relevant reviews (e.g., Serre, 2019) and work showing the influence of architectural constraints on the interpretation of network response patterns (e.g., Revsine et al., 2024).

Thank you. As detailed in response to major concern 11, we included a full paragraph on this issue in the discussion (including discussion of Revsine et al., 2024 and many more papers). In addition, we expanded the initial exposition of the use of DNNs as follows:

“To probe this developmental account of the emergence of the parvo- and magnocellular systems, we conducted simulations with deep convolutional neural networks. While these powerful networks are not without their limitations as computational models of the biological system (reviewed in Serre, 2019; Wichmann & Geirhos, 2023; Bowers et al., 2023), they are among the best to predict human behavior and neural responses (e.g., Lindsay, 2021; Schrimpf et al., 2020; Storrs et al., 2021). Moreover, they offer a framework for scientific exploration and the testing of neuroscientific hypotheses (Cichy & Kaiser, 2019; Doerig et al., 2023). Of particular relevance to us, they provide a systematic methodology for directly probing the consequences of deliberate manipulations of early sensory inputs (Vogelsang & Sinha, 2023). Unlike in human participants, where such manipulations are not feasible for obvious ethical reasons, we can expose a deep network to different temporal progressions of inputs and analyze the effect on learned representations and performance outcomes.”

2) Terminology line 69: manipulation of sensory experience or do the authors mean sensory input?

We changed it to “input”.

3) Lines 107-109: Statistical test may be unwarranted (strongly skewed distributions)

Thank you for raising this point. In light of the skewed distributions, we have removed the t-test.

4) Line 125: “[...] we observe a clear cluster of magnocellular-like receptive fields [...]” The claim that there is a cluster has not been substantiated by objective criteria. It would be also important to clarify in what sense these RFs are not M-like.

As explained in response to the first major concern in the Methods section, we have changed our phrasing throughout the manuscript. In addition, we now mention the non-discreteness of distributions in the Discussion section, as follows:

“Moreover, while biomimetic training yields a population of magnocellular-like units and more heterogeneous parvocellular-like units, these distributions are not categorical and may represent a broader continuum.”

5) Figure 2C: what is different about cars & boats?

Thank you for bringing up this point. What differentiates classes with more vs. less biomimetic-to-standard-benefits is an interesting question. It is important to note that, based on the helpful suggestion of reviewer 2, we now re-ran this analysis (and other analyses) on the basis of five different training runs with different random initializations. This provided us with a clearer view on the stability of any class-specific differences. As is evident in revised Figure 2C (included below), when taking the mean of the 5 training runs, there are now zero classes (rather than two classes: boats and cars) that have a greater shape bias in the standard model, compared to the biomimetic one:

Figure 2C. Percentage of shape-based correct classifications, as opposed to texture-based correct classifications, for each of the 16 different super-classes. Classifications that were inconsistent with both texture and shape classes are not included in this computation. Shown here are results of the five individual training runs (dots), along with their means (bars). Depicted on the y-axis are shape categories.

One could point out that the class for which both models classify most shape-like is featuring a clock. Quite similarly, for humans (as shown in Geirhos et al., 2018), this is the second-most shape-like class. On the other end of the spectrum, airplanes are the most texture-based class for both models. For humans, quite similarly, this corresponds to the third-most texture-based class. Thus, it is possible that the class-by-class differences could be explained by the relative importance of texture and shape information. However, we did not include any speculation about specific class-by-class differences in the manuscript. Instead, we state that:

“While there are some differences across the 16 different classes, the results of both models are overall fairly consistent across classes.”

6) It may make sense to change x-axes in Figure 3 to: Correct decisions (%), and indicate chance-level in figures, as well as construct & report confidence intervals.

Thank you for this comment, which prompted us to refine our axis labels. As for Figure 3 specifically, the x-axes describe the proportion of ablated receptive fields (Figures 3A&B) as well as correlation values (Figures 3C&D), and we were not sure in what context to incorporate “Correct decisions (%)”. However, if the comment might have been about the y-axis label and the inclusion of “%” as unit, we, in fact, renamed “Test performance (gray)” and “Test performance (color)” to “Performance on gray (%)” and “Performance on color (%)” for Figures 3A&B. Moreover, we have also changed the x-axis labels from “Nr of RFs ablated” to “RFs ablated (%)” for Figures 3A&B (and Figures 2D&C). We also renamed the x-axis label of Figure 2C to “Shape-based, not texture-based, correct decisions (%)”. However, we kept the y-label “Total classification” as y-labels in Figures 2B/D/E to differentiate it from Figure 2C.

Thank you also for your comment about chance levels. In Figure 3A&B, as there is a total of 1,000 ImageNet classes, chance level would be at 0.1% and not visible if plotted. Therefore, we added it to the figure caption:

“As there are 1,000 ImageNet classes, the chance level is 0.1%.”

In Figure 2, we have now added a dashed line at 1/16, as a total of 16 classes underlie the shape/texture bias methodology by Geirhos et al. (2018). We added the following to the figure caption:

“The dashed line, plotted at 1/16, indicates the chance classification level expected of a network generating random responses.”

Regarding the point of confidence intervals: as alluded to above, based on reviewer 2’s helpful suggestions, we re-ran the simulations five times and report results of all five runs. In Figure 2B, we are reporting standard errors along with individual data. In the other panels, we are reporting means along with individual data.

7) It would be helpful to read more (perhaps in the SI) about how the authors impute decisions between shape-based/not texture based, etc.

We have significantly expanded the Methods section description related to the texture-shape analysis. The revised section reads as follows:

“For the results reported in Figure 2 (and corresponding Figures S13-18), we present a texture/shape bias analysis using the dataset introduced by Geirhos et al. (2018). Their

test images are publicly available at: <https://github.com/rgeirhos/texture-vs-shape/tree/master/stimuli/style-transfer-preprocessed-512> (more details about the stimuli and methodology can be found in Geirhos et al., 2018). This dataset contains a total of 1,280 images in which global shape and local texture cues are in conflict. For instance, one image might exhibit the global shape of a cat but the local texture of an elephant. If a network classifies such an image as a cat, its decision would be deemed shape-consistent. If, on the other hand, the network labels such an image as an elephant, it would be judged as texture-consistent. Classifications that do not match either of these categories are considered incorrect (depicted as ‘neither’ in Figure 2).

While ImageNet encompasses a total of 1,000 image classes, the shape-texture methodology consolidates a subset of these into 16 broader, relatively high-level categories (listed on the y-axis of Figure 2C), each representing a superset of the original ImageNet classes (the mapping can be found in the original repository by Geirhos et al., 2018: https://github.com/rgeirhos/texture-vs-shape/blob/master/code/helper/human_categories.py). The 1,280 images were generated such that there are 5 images for each possible pairing of these 16 shape classes with the same 16 texture classes. Due to this design, 80 of the 1,280 images include matching shape and texture cues. Following the analysis methodology used in the original paper, we excluded those 80 images from our shape-texture analysis.

For each of the relevant 1,200 images, we obtained the network’s final-layer predictions (1,000 probabilities, one per ImageNet class). Because not all of the 1,000 ImageNet classes map onto the 16 broader categories, we restricted our analysis to only those classes that do. We selected the class with the highest probability from this subset and mapped it to one of the corresponding 16 categories. Based on whether the chosen class matched the shape or texture of the shape-texture conflict image, the network’s response was labeled shape-consistent, texture-consistent, or incorrect (if the class belonged to one of the remaining 14 categories). Note that in Figure 2C, results are shown for each shape category separately. Specifically, for each of the 16 shape categories, we show the models’ responses to the 75 images (1,200 total images / 16 categories) in which the corresponding shape was present.”

8) Terminology: to my understanding RF’s are not ablated. Units (or neurons) with RFs are ablated. E.g., see Figure 3 axes & text.

We have now clarified upon the first use of the term ‘receptive fields’ in the Results section, that it corresponds to the learned convolutional kernels:

“Figures 1A&B depict the learned convolutional filters (henceforth, ‘receptive fields’) in the first layer of our network (...)”

Moreover, we added the following clarification to the Results section:

“The ablation of each receptive field involves ablation of all of its (here, 22x22x3) values.”

9) Terminology: Caption Fig 3 line 204: colorful first-layer receptive fields. Perhaps color-sensitive works better?

We have now rephrased ‘colorful’ with ‘color-sensitive’ throughout the Results section.

10) Line 216: what does this mean? “Indicative of the magno stream’s greater temporal dynamics”

Thank you for pointing out this potentially unclear phrasing and explanation. We have made two changes in response. First, in the Introduction section, we have expanded the exposition of temporal receptive field properties as follows:

“In addition to these spatial characteristics, the two pathways also differ in their temporal response properties. Magno cells are known to exhibit faster conductance (Schiller & Malpeli, 1978; Dreher et al., 1976) and higher temporal frequency sensitivity (Derrington & Lennie, 1984; Usrey & Reid, 2000; O’Keefe et al., 1998). Specifically, the temporal frequencies eliciting maximal activation were found to be approximately twice as high in magnocellular neurons in the monkey LGN, relative to parvocellular ones (Derrington & Lennie, 1984; Usrey & Reid). Furthermore, De Valois et al. (2000) showed that the magnocellular system includes a greater proportion of biphasic cells, which reverse their phase responses over time and thereby enable the encoding of rapid temporal transients.”

Second, we changed the phrasing from “indicative of the magno stream’s greater temporal dynamics” to:

“allowing the magno stream to encode more rapid temporal transients”

11) The authors focus on finding similarities between the literature and their observations. It would be probably good to focus a bit more on differences.

Thank you for this important suggestion. As detailed in response to major concern 12, we added a third limitation paragraph concerned with differences between the receptive fields found in our study and those of the real biological system. In addition, as detailed in response to major concern 15, we have included another paragraph related to observations that challenge the notion of magnocellular susceptibility to early deprivation.

12) Line 295-302: In my view this speculation needs to be further clarified. It may be useful to distinguish optical properties from neural properties given the aims of this study.

As detailed in response to major concern 15, we have reworked the entire section. The specific part relating to the comment reads:

“As many of the maturational processes, which are significantly driven by retinal development (Banks & Crowell, 1993), continue despite the children’s blindness, their post-surgical visual system is markedly more mature than that of a newborn (Banks and Crowell, 1993, Wilson, 1993, Boas et al., 1969, Hendrickson and Boothe, 1976). Indeed, their behaviorally-assessed visual acuity, while below that of typically developed controls, is above neonate levels (Ganesh et al., 2014), and their color sensitivity reaches normal levels even just a few days post-surgery (Vogelsang et al., 2024a).”

Thus, it is not just the optical properties but even evident at the behavioral level (and thus, necessarily, at the neural level).

13) I am not sure what is the benefit of providing teleological perspectives in this context. Please clarify.

Thank you for raising this point. Instead of focusing on teleology, we have rephrased the sentence to reflect the adaptiveness and potential functional significance. The sentence now reads:

“In conclusion, the work presented here offers a potential account for the emergence of the parvo/magno distinction and thereby also provides a framework for understanding the potential adaptive significance of why normal development progresses in a particular temporal sequence.”

We would like to thank the reviewer again for the many helpful suggestions, which have significantly improved our paper.

Reviewer #2 (Remarks to the Author):

Strong Points:

- * Innovative approach using multiple training regimes, including mixtures of blurred and color inputs.
- * Highly relevant and insightful analysis of receptive field properties in DCNN units, with a notable inclusion of motion studies.

We thank the reviewer for highlighting these strengths of our manuscript!

Minor Points:

- * reference work on k-cells (e.g., VA Casagrande, TINS 1994), which might also enrich your discussion.

Thank you for this important suggestion, which further strengthens the discussion of the matter. We have added the following to the Discussion section:

“Finally, our analyses do not explicitly consider koniocellular (K) cells, which constitute a highly heterogeneous collection in the primate LGN and feature multiple subclasses of units with distinct response properties (reviewed in Casagrande, 1994 and Hendry & Reid, 2000). As Hendry & Reid (2000) note, the existence of this additional and highly heterogeneous pathway, alongside the parvo- and magnocellular streams, “introduces a complexity into a field only recently viewed as clean and simple”. Thus, while our approach captures some aspects of the organization in the mammalian visual system, a more comprehensive account will need to incorporate this additional complexity as well.”

- * Clarify the rationale for using 48 receptive fields of 22x22 in layer 1 instead of the default network configuration, especially if results are similar. Why deviate from the standard?

Thank you for highlighting the need to clarify the rationale of these two architectural adjustments. The decision to increase the receptive field size from 11x11 to 22x22 pixels was driven by the desire to be able to extract frequency-based metrics with higher resolution and precision, and wanting to avoid restricting the types of receptive field structures that can be learned by the network (having larger first-layer kernels allows, in principle, learning to integrate across larger spatial extents). The decision to reduce the total number of first-layer receptive fields from 96 to 48 sought to avoid potential over-parametrization and/or the presence of receptive fields that only exhibit weak structures and primarily represent noise. Specifically, as also detailed in response to reviewer 1, before training, receptive field weights are randomly initiated. Then, throughout training, receptive fields tend to specialize and form clear structures, which can be meaningfully and reliably characterized through the use of metrics. If certain receptive fields were to not learn any clear structures but remain very noisy, the reliability of analyses may be reduced. Therefore, as a precaution, our first approach was to reduce the number of receptive fields (also considering that we had previously increased the receptive field sizes from 11x11 to 22x22 pixels). With this smaller set of receptive fields to be learned, we expected that there would be an even stronger need for the network to build meaningful structures. Indeed, although from a different domain, in the context of learning genomic

sequence motifs, past work by Koo & Eddy (2019) has shown that the use of too many first-layer convolutional filters can lead to more filters not learning any meaningful representations.

Based on your very helpful comment, we have revised the manuscript in two important ways. First, we are reporting, in greater detail and with each simulation repeated five times, results with (i) 48 receptive fields of size 22x22, (ii) 96 receptive fields of size 22x22, and, now also, (iii) 96 receptive fields of size 11x11. For direct visual comparison, included below is a compilation of Figure 1E&F, Supplementary Figure 3E&F, and Supplementary Figure 5E&F, each depicting the receptive field analysis resulting from the first training run. (Outcomes of all five training runs, revealing qualitatively similar results are shown in Supplementary Figures 2, 4, and 6. For texture/shape results of settings 2 and 3, see Supplementary Figures 13 & 14.)

Compilation of Figures 1, S3, and S5: Scatter plots depicting the joint frequency and color coding of individual RFs following training with the standard (left) and biomimetic regimen (right) for setting 1 (top), setting 2 (center), and setting 3 (bottom).

As is evident from this figure, the basic patterns observed hold across all different settings tested.

Second, and relatedly, we have now explained our architectural adjustment in the Introduction section. In this context, we have also highlighted that many other simulations were run (including those with the original set of 96 11x11 pixel receptive fields), and used a different

phrasing to reflect that simulations across all of these settings are important. The revised Introduction paragraph reads as follows:

“To examine the generalizability of our results, these comparisons were carried out across several network settings, which are illustrated in Figure S1 in the supplementary material. In ‘setting 1’, for which results are reported in the main manuscript, we adapted the AlexNet to feature larger first-layer filters (increased from 11x11 to 22x22 pixels) in order to avoid restricting the types of receptive field structures to be learned and to allow for the extraction of frequency-based metrics with higher resolution. We also reduced the number of receptive fields from 96 to 48 as a precautionary measure to increase the likelihood of all receptive fields learning clear structures that can be meaningfully quantified through the use of metrics. In addition, we also report results with the full set of 96 receptive fields (size increased to 22x22 pixels) (‘setting 2’), with the full set of 96 normally-sized (i.e., 11x11 pixel) receptive fields (‘setting 3’) (...)”

In addition, we added to the Methods section:

“For setting 1 reported in the main manuscript, we slightly adapted the original architecture on two fronts. First, we enlarged the size of the receptive fields in the first convolutional layer from 11x11 to 22x22 pixels in order to avoid restricting the types of receptive field structures that can be learned by the network, and to allow for the extraction of frequency-based metrics with higher resolution and precision. Second, the number of first-layer receptive fields was reduced from a total of 96 to 48 as a precautionary measure to increase the likelihood of all receptive fields learning clear structures that can be meaningfully and reliably quantified through the use of metrics. (Although originating from a different domain, in the context of learning genomic sequence motifs, past work has shown that too many first-layer convolutional filters can lead to more filters not learning meaningful representations (Koo & Eddy, 2019).) (...).”

as well as:

“For setting 2 reported in the supplementary material accompanying this paper, we built on setting 1 but used the full set of 96 first-layer receptive fields (when enlarged to 22x22 pixels).

For setting 3, we built on setting 1 but used the original number and size of receptive fields in the first layer (i.e., a total of 96 receptive fields, each of size 11x11 pixels). This setting was included to verify whether even with the original architecture (not optimized for fine-grained frequency-based assessments of individual receptive fields), the basic results would hold. “

* What is the next convolutional layer doing? Could that be relevant?

Thank you for raising this interesting point. A key observation we made in the first convolutional layer is that the biomimetic, but not the standard, model exhibits a population of magnocellular-like receptive fields that are *jointly* tuned to low spatial frequency and low chromatic information. Due to complex connectivity patterns between the first convolutional and deeper layers (and, similarly, in the biological system, a reported mixing of the cortical pathways; Hubel & Livingstone, 1990), one would expect a reduction of any ‘joint’ coding. However, examining these two dimensions independently, one might expect that the

biomimetic model maintains tuning to lower spatial frequencies and chromatic content throughout the network and, consequently, invariance of activations to color removal or blur. Indeed, this is what we observe in our invariance analysis depicted in Figures 3C&D. To demonstrate this point more clearly, we have now extended this analysis (now, like all other analyses, also based on the repetition of 5 training instances) to all network layers. This figure, included below, reveals that across all network layers, the biomimetic model indeed maintains greater robustness to chromatic and blur changes:

Figure 3C&D. Distribution of correlations of neural activations across layers between full-color and grayscale images (C) and between full-frequency and blurred images (D). Depicted are superimposed results of the five individual network training runs.

We would like to thank you again for bringing up this point, also because it made us realize that there had been an error in the layer names as y-labels, which we have now corrected.

Moreover, we added the following interpretation to the Results section:

“For both stimulus dimensions, the obtained distributions lean towards higher correlation values for the biomimetic than for the standard model, suggesting greater invariance of the biomimetic model. Notably, this is the case across all network layers. Thus, while we previously focused on the first convolutional layer for analyses of the joint coding of spatial frequency and color sensitivity in individual receptive fields, some of the biases in terms of color and spatial frequency encoding appear to be maintained in deeper network layers as well.”

* Explain the choice of 200 epochs with a constant learning rate. Why? I do not think this is the SOTA training for Alexnet.

Thank you for this suggestion. The primary motivation for using a constant, rather than decreasing, learning rate was to render the number of epochs more interpretable, facilitate simple replicability, and ensure that any potential benefits from the first half of training in the biomimetic model are not just maintained due to effectively too low learning rates in the second half. However, you raise an important point, in response to which we have now also included simulations with decreasing learning rates (which we refer to as ‘setting 5’). With this setting, we

obtain qualitatively similar results. Included below is Supplementary Figure 9E&F (for texture/shape results with this setting, see Supplementary Figure 16):

Fig S9. Reproduction of Figure 1 in the main manuscript when utilizing setting 5 (decreasing learning rate). **E&F.** Scatter plots depicting the joint frequency and color coding of individual RFs. Depicted here are results obtained with the first training run; outcomes of five training runs with different random initializations are shown in Figure S10.

Moreover, our decision to feature a relatively large number of epochs (a total of 200) in our constant learning rate setting was driven by the desire to have clear convergence for all training regimens and to lead to clear filters in the first convolutional layer, which may be finetuned even when performance has (almost) converged. However, for comparison, we now also ran the simulation with shorter epochs (a total of 100 epochs with constant learning rate; which we refer to as ‘setting 4’). Again, the results are relatively stable. Included below is Supplementary Figure 7E&F (for texture/shape results with this setting, see Supplementary Figure 15):

Fig S7. Reproduction of Figure 1 in the main manuscript when utilizing setting 4 (fewer epochs). **E&F.** Scatter plots depicting the joint frequency and color coding of individual RFs. Depicted here are results obtained with the first training run; outcomes of five training runs with different random initializations are shown in Figure S8.

We added to the Introduction section:

“In addition, we also report results with (...) only half of the training epochs (‘setting 4’), and with a decreasing learning rate (‘setting 5’). ”

We also added to the Methods section, in the context of justifying the training parameters for ‘setting 1’:

“(...) Training lasted for a total of 200 epochs, and a constant learning rate was used. These settings were kept deliberately simple (rather than featuring, for instance, complex decreasing learning rate schedules) in order to render the number of epochs more interpretable, facilitate simple replicability, and ensure that any potentially beneficial effects emerging from the first half of training in the biomimetic model are not just maintained due to effectively too low learning rates in the second half. Moreover, we trained for a relatively extended time to ensure fair convergence of all models (see Figure S21 for depiction of loss and accuracy throughout training) and to be able to obtain well-established receptive field structures that can be analyzed meaningfully and reliably. However, to examine the generalizability of our findings, we systematically varied a few of these parameters, as described below.”

as well as for settings 4 and 5:

“For setting 4, we used setting 1 but only trained for a total of 100, instead of 200, epochs. This setting was included to ensure that the observed findings are not just the consequence of a comparatively long total training duration.

Finally, for setting 5, we used setting 1 but, to examine generalizability to more modern training procedures, implemented a decreasing learning rate schedule (for details, see below).”

Finally, note that, in order to provide an overview of the 5 different network settings, we included a schematic as Supplementary Figure 1.

Fig. S1. Illustration of the different settings and regimens used for training our 2D networks.

* Include visualizations of loss functions for all networks to aid comparison.

Thank you. We have now included visualizations of training / validation loss and accuracy for all models tested as Supplementary Figure 21, shown below.

Fig S21. Depiction of training and validation loss / accuracy across the different settings and regimens used for training our 2D networks. Depicted are superimposed results of the five individual network training runs.

As is evident, the models do show fair convergence while not yet exhibiting overfitting. Related to the point raised above, 200 epochs with constant learning rate are primarily used to ensure good convergence of the receptive field structures in the first layer. Results with 100 epochs with constant learning rate, and results with 200 epochs with decreasing learning rate, are relatively similar.

Major points:

* Robustness of results: Only one trained version of each network is presented. To establish reliability and rule out random drift, it is crucial to train multiple versions (e.g., 5) with different initializations. Present means with standard errors.

Thank you for bringing up this critical point. We have now repeated the training of all models in all settings with 5 different random initializations. For Figures 2 and 3, we incorporated results into a single figure (always featuring individual data as well). They are included below and reveal the consistency of results across training runs:

Figure 2. A. Exemplar shape/texture cue conflict stimulus, reprinted from Geirhos et al. (2018a), depicting the local texture of an elephant and the global shape of a cat. **B.** Percentage of total classifications correct in terms of shape, correct in terms of texture, or neither (i.e., incorrect). Error bars represent the standard error across the five different training runs with random initializations, and dots depict results of each individual run. **C.** Percentage of shape-based correct classifications, as opposed to texture-based correct classifications, for each of the 16 different super-classes. Classifications that were inconsistent with both texture and shape classes are not included in this computation. Shown here are results of the five individual training runs (dots), along with their means (bars). Depicted on the y-axis are shape categories. **D&E.** Shape/texture bias as a function of the proportion of the ablated least color-sensitive (left) and most color-sensitive (right) units, for the standard (D) and biomimetic (E) regimen. Depicted are the five training runs (thin lines) as well as their means (thick lines). The dashed line, plotted at 1/16, indicates the chance classification level expected of a network generating random responses.

Figure 3. A&B. Classification performance on color (top) and grayscale (bottom) images when ablating the least vs. most color-sensitive (A) as well as least vs. most high spatial frequency tuned (B) first-layer receptive fields. Depicted are results of five training runs with different random initializations (thin lines), along with their means (thick lines). As there are 1,000 ImageNet classes, the chance level is 0.1%. **C&D.** Distribution of correlations of neural activations across layers between full-color and grayscale images (C) and between full-frequency and blurred images (D). Depicted are superimposed results of the five individual network training runs.

For receptive field plots, we did not merge the results across the five training runs but depict each separately. For the 2D-CNNs in the standard setting (setting 1), the results are included as Supplementary Figure 2:

Fig. S2. Depiction of individual and joint metric distributions for setting 1 (48 22x22 pixel RFs) across all five training runs with different random initializations (the first run is shown in Figure 1 in the main manuscript). **A&B.** Color and spatial frequency distributions of individual RFs. **C&D.** Scatter plots depicting the joint frequency and color coding of individual RFs.

For the 3D-CNN, the results are included as Supplementary Figure 20:

Fig S20. Depiction of joint RF metric distributions of our 3D CNNs across all five training runs with different random initializations (the first run shown in Figures 4C & 4F). Depicted is the relationship between temporal RF properties (using the temporal variation metric) and spatial RF characteristics (using the color and spatial frequency metrics used before) of the both the biomimetic and standard model.

Again, results are fairly stable across random training runs.

* Discuss the methods to characterise the units (color, orientation, RF size) in relation to existing literature on this topic. How does this approach compare.

Thank you for raising this important point. Indeed, our receptive field metrics are based on past physiological and computational work, but were adapted to the specific study context.

In neurophysiological experiments, the approach is typically to measure a neuron's responses while systematically varying a dimension of the stimulus, such as its orientation, spatial frequency, or color. In addition to a depiction of the measured responses as a function of the varied parameter of interest, the responses can be analyzed (and thereby summarized to a single value) using defined metrics. For the receptive field computations reported in this paper, we have not measured responses to stimuli but analyzed the learned kernels directly. Nevertheless, the analyses applied to the learned filters have been inspired by past physiological work, as detailed below:

1) Orientation tuning: In our simulations, we applied a 2D FFT to a given filter (after converting it to grayscale), and then averaged across frequencies (through azimuthal averaging) to obtain amplitude as a function of orientation (ranging from 0 to π). This is stored as a 1-dimensional vector. Then, using the mean resultant length, calculating the variance in circular space, we captured how strongly a given filter is tuned to a specific orientation. This computation is directly inspired by the 'circular variance' metric established in neurophysiology (see Ringach et al., 2002, *Journal of Neuroscience*), where, similarly, the resultant is applied to orientation tuning curves (also a 1-dimensional vector as a function of orientation), in order to extract how orientation-selective a given cell is. This measure has been used in several later studies (e.g., Shushruth et al., 2013; *Journal of Neuroscience*). Thus, while physiological studies differ from our computational one in that actual firing is observed as a function of orientation, rather than the result of an FFT, the resulting orientation vectors have been analyzed in comparable ways.

We added the following to the Methods section:

“This computation is inspired by the ‘circular variance’ metric widely used in neurophysiological studies, where, similarly, the resultant is applied to 1D tuning curves as a function of orientation in order to extract how orientation-selective a given cell is (Ringach et al., 2002).”

2) Spatial frequency metric: To quantify spatial frequency sensitivity, we applied a 2D-FFT to a given filter (after converting it to grayscale) and then computed the radial average, resulting in a 1D vector as a function of frequencies. Given this vector, we compute, as detailed in the Methods, the weighted average frequency:

$$\text{weighted average frequency} = \frac{\sum_f \text{amp}(f) * f}{\sum_f \text{amp}(f)}$$

Similar to the case of orientation selectivity, neurophysiological studies would measure a given neuron's firing across different spatial frequencies, resulting directly (rather than after applying an FFT and radial averaging to a filter) in a tuning curve (i.e., a 1D vector) as a function of frequency. Reflecting the notion of 'preferred stimuli' that excite a given neuron maximally, a common approach in neurophysiology is to characterize this tuning curve by extracting the frequency that elicits maximal firing (e.g., O Keefe et al., 1998; Bredfeldt & Ringach, 2002).

In a hypothetical scenario where a neuron fires exclusively for a certain frequency and does not fire otherwise (i.e., the vector is zero for all other entries), the frequency eliciting maximal activity would be the same as the one extracted using our weighted average frequency metric. In more realistic scenarios, the difference is that our metric also takes into account the firing to other frequencies (in the neurophysiological context) / the presence of the non-maximal spatial frequencies (in the computational context) to allow for a more fine-grained characterization of individual receptive fields' spatial frequency tuning. In our case, this is especially pertinent as the resulting values would be fairly discrete when extracting only the frequency of highest amplitude. This is due to the small size of the receptive field (even if increased to 22x22 pixels and up-sampled prior to application of the FFT). Moreover, given the typically strong decay of the FFT response as a function of frequency, most filters would be evaluated at the same, fairly low, value. The metric, thus, would not capture the fine-grained differences between individual receptive fields, which would not allow for the subsequently carried out analyses.

To illustrate this point, we computed this alternative metric (i.e., the frequency of highest amplitude) and compared it to our weighted average frequency metric. Specifically, we took individual RFs of the trained networks for 'setting 1' reported in the main manuscript and plotted individual values of the alternative metric (y-axis) against those resulting from the weighted average frequency metric used in the manuscript (x-axis). To facilitate visual inspection, we merged the data across the 5 random initializations. The scatter plot, thus, contains $5 \times 48 = 240$ dots for the biomimetic and 240 dots for the standard regimen.

While for many of the RFs tuned to very high frequencies, the dots are close to the diagonal (i.e., the extracted metric values are similar), most of the receptive fields are evaluated as having similar, and fairly low, values with the alternative metric. Using such measure, one can still see that the biomimetic model is tuned to lower frequencies than the standard one, and one would likely be able to find an anti-correlation between color and spatial frequency tuning for some of the RFs in the standard model. However, this metric would render any joint examination of the biomimetic model unreliable as it is not sensitive enough and almost all receptive fields are at the minimum.

We added the following to the Methods section:

“Finally, note that in neurophysiological studies, a common technique to summarize spatial frequency tuning curves (i.e., a similarly 1-dimensional vector as a function of frequency) is to simply extract the frequency eliciting maximal activity (see, e.g., O Keefe et al., 1998 or Bredfeldt & Ringach, 2002). Here, we chose to use the weighted average frequency metric in order to not just describe the frequency corresponding to maximal amplitude but to also take into account the amplitudes of all other frequencies. This is especially important given the rather small size of RFs (even if increased to 22x22 and up-sampled prior to the application of the FFT). Overall, for the specific characterization carried out here, this approach yields a more sensitive and fine-grained differentiation between individual receptive fields and prevents clustering around fairly discrete values.”

3) Color sensitivity metric: There are two steps to our color sensitivity metric. First, we quantify the colorfulness of each pixel in a given receptive field (using RGB color space). Then, we summarize the pixel-wise values by taking the mean of the approximately 10% most colorful pixels. We now provide more context for this consideration, and then comment on both of the above analysis decisions.

Prior to describing our color metric, we have now added the following to the Methods section for context:

“Physiological studies have often characterized neurons in terms of what specific colors or color opponencies they are most sensitive to. Of greater relevance, other measures have been established in neurophysiology (Conway et al., 2007), which were subsequently used in computational modeling studies (Flachot & Gegenfurtner, 2018), that characterize the overall sensitivity/responsivity to color based on the strength of responses to color vs. grayscale stimuli. In our case, however, working with learned filters rather than recorded responses, and allowing for the analysis of colorfulness of sub-sections of the receptive field (to not introduce biases in estimating spatially-extended receptive fields as more colorful), we quantified the color sensitivity (or colorfulness) of a given receptive field in two steps.”

Following the description of our color metric, we added:

“While we have chosen the RGB color space to characterize deviations from grayscale (by quantifying imbalances between R, G, and B values), one could have done so in other color spaces as well. Figure S22 shows the results (for setting 1) when extracting the saturation (S) in HSV space or chroma ($\sqrt{a^2+b^2}$) in CIELAB color space instead. Overall, qualitatively similar results hold.”

We include Supplementary Figure 22 below. It shows that quantifying color imbalances in RGB color space does not yield markedly different results from doing so in other color spaces. Specifically, comparing the color-sensitivity metric results (of setting 1 reported in the main manuscript) when using HSV color space (panels A and D) or CIELAB color space (panels B and E), as opposed to RGB color space (x-axes of panels A and B; and panel C), qualitatively similar results hold: biomimetic training leads to receptive fields with lower color tuning, lower spatial frequency tuning, as well as jointly lower color and spatial frequency tuning.

Fig S22. Further examination of our color metric. **A&B.** Relationship between individual RF characteristics based on our color metric (x -axis) and alternative color metrics based on the HSV and CIELAB color spaces (y -axes). Depicted are data corresponding to individual receptive fields across the 5 individual training runs for our main setting (setting 1). **C-E.** Scatter plots depicting the joint frequency and color coding of individual RFs, when using the original RGB-based, the HSV-based, or the CIELAB-based metric.

Finally, we described in the Methods section (now also revised to adjust the metric for setting 3 appropriately due to receptive fields having size 11x11 pixels):

“We then summarized the distribution of such color channel discrepancies across the 22x22 pixels of a given receptive field into a single value. Defining our final color metric as the mean of the 22x22 distribution would induce an estimation bias towards spatially extended color receptive fields. Instead, taking into account only the single most colorful pixel could potentially be subject to high noise. As a compromise, we chose to define our final metric as the average of the top-48 (for settings 1, 2, 4, and 5; the top-12 for setting 3; i.e., approximately the top-10%) most colorful pixels within a given receptive field – roughly approximating the size of the smallest effective receptive fields.”

We also added to this paragraph:

“Note, however, that qualitatively similar results hold when taking even fewer of the most colorful pixels into account.”

This is illustrated in the additional plot below. Comparing the color-sensitivity metric results (of setting 1 reported in the main manuscript) when using the top-12 (panels A and C) or the top-1 (panels B and D), as opposed to the top-48, most colorful pixels, again, qualitatively similar results hold: biomimetic training leads to receptive fields with lower color tuning, lower spatial frequency tuning, as well as jointly lower color and spatial frequency tuning.

* Discuss how this work relates to other papers looking at smoothing of stimuli used for training (for instance Frank Tong's work).

Thank you for this important suggestion. Along with a much more thorough literature both in the Introduction and Discussion sections (as detailed also in responses to reviewer 1), we now discuss this specific relation up-front in the Introduction:

“In the domain of visual acuity, Vogelsang et al. (2018) have shown that training deep convolutional neural networks with inputs that transition from blurry to non-blurry leads to first-layer receptive fields tuned to lower spatial frequencies, effectively integrating information over more extended areas of the input image. This training progression, compared to several non-developmental ones, also yielded the best generalization performance across varying blur levels, suggesting potential benefits of commencing training with blurry inputs, at least in the domain of face recognition (Vogelsang et al., 2018; Jang & Tong, 2021). Although without an explicit developmental motivation, Jang & Tong (2024) recently also demonstrated that training a deep network with various levels of blurred inputs (including strong blur) can lead to a greater similarity with neural responses in macaque visual cortices, greater robustness to image perturbations, and an increased bias to classify images based on global shape rather than local features. Yoshihara et al. (2023) also report a slightly increased shape bias, but more modest than the strong-blur results reported by Jang & Tong (2024), when training with a mix of blurred and non-blurred images or when transitioning from blurred to non-blurred inputs.”

In addition, we have added the following to the Discussion section:

“Our study also connects with and adds to recent work by Jang & Tong (2024), who reported an increased shape bias when networks were trained with a mix of blurred inputs (including strongly blurred ones). Specifically, given the proposed linkage between receptive field properties and global shape biases, we present a hypothesis about how such shape biases might arise at the representational level. Moreover, we demonstrate that increased shape biases can also emerge in developmentally plausible settings (that is, for progressions transitioning from poor to rich), and we have also incorporated joint trajectories in both color and spatial frequency.”

We would again like to thank the reviewer for the many helpful suggestions, which have significantly improved our paper.

Responses to reviewer comments

Reviewers' comments:

Reviewer #2 (Remarks to the Author):

I want to thank the authors for their very extensive, and adequate replies to my queries. With the extensive work done for me (reviewer 2) and also looking at the work that has been done for reviewer 1 I think the paper improved in quality substantially. The first version of the paper was a joy to read and this has only improved.

We would like to thank the reviewer again for the many helpful suggestions, which have significantly improved our study. We are glad to hear about the positive assessment of this work and the revised version of this manuscript.

I am left with only 1, more minor, question. that should be commented upon somewhere in the paper: as expected from the loss in Fig S21 the biomimetic networks take longer to reach a solid performance level. While not unexpected (and explaining the 200 training epochs) this is important to comment on.

We have now included the following in the Discussion section: *“In interpreting comparative benefits of the biomimetic vs. standard training regimens, it is important to acknowledge, however, that biomimetic training requires more epochs to achieve stable performance levels, as indicated by slower convergence of training loss (Supplementary Figure 21).”*

We have also added the following to the Methods: *“To ensure fair and stable convergence of all models, we employed a relatively extended number of epochs. It is worth noting that biomimetic models require more epochs to reach stable levels of loss and accuracy, as illustrated in Supplementary Figure 21.”*